# A common alteration in effort-based decision-making in apathy, anhedonia, and late circadian rhythm

Sara Z Mehrhof[1]*, Camilla L Nord[1,2]

[1]MRC Cognition and Brain Sciences Unit, University of Cambridge, Cambridge, United Kingdom; [2]Department of Psychiatry, University of Cambridge, Cambridge, United Kingdom

## eLife assessment

This **important** study provides **convincing** evidence that both psychiatric dimensions (e.g. anhedonia, apathy, or depression) and chronotype (i.e., being a morning or evening person) influence effort-based decision-making. This is of importance to researchers and clinicians alike, who may make inferences about behaviour and cognition without taking into account whether the individual may be tested or observed out-of-sync with their phenotype. The current study can serve as a starting point for more targeted investigation of the relationship between chronotype, altered decision making and psychiatric illness.

**\*For correspondence:**
sara.mehrhof@mrc-cbu.cam.ac.uk

**Competing interest:** The authors declare that no competing interests exist.

**Abstract** Motivational deficits are common in several brain disorders, and motivational syndromes like apathy and anhedonia predict worse outcomes. Disrupted effort-based decision-making may represent a neurobiological underpinning of motivational deficits, shared across neuropsychiatric disorders. We measured effort-based decision-making in 994 participants using a gamified online task, combined with computational modelling, and validated offline for test–retest reliability. In two pre-registered studies, we first replicated studies linking impaired effort-based decision-making to neuropsychiatric syndromes, taking both a transdiagnostic and a diagnostic-criteria approach. Next, testing participants with *early* and *late* circadian rhythms in the morning and evening, we find circadian rhythm interacts with time-of-testing to produce parallel effects on effort-based decision-making. Circadian rhythm may be an important variable in computational psychiatry, decreasing reliability or distorting results when left unaccounted for. Disentangling effects of neuropsychiatric syndromes and circadian rhythm on effort-based decision-making will be essential to understand motivational pathologies and to develop tailored clinical interventions.

## Introduction

Our circadian rhythm aligns us with our environment, regulating physiological and behavioural processes to follow 24-hr rhythms (*Foster, 2020*). Circadian integrity is pivotal to mental wellbeing and has been bidirectionally linked to numerous psychiatric disorders (*McCarthy, 2019*; *Logan and McClung, 2019*; *Walker et al., 2020*; *Ashton and Jagannath, 2020*; *Lyall et al., 2018*; *Vadnie and McClung, 2017*; *Emens et al., 2009*). Yet little is known about the cognitive or computational mechanisms of circadian dysfunction—and their alignment or diversion from mechanisms driving neuropsychiatric symptoms.

Inter-individual differences in circadian timing and alignment manifest behaviourally as chronotypes (i.e., diurnal preference) (*Jones et al., 2019*; *Bailey and Heitkemper, 2001*), with individuals

**eLife digest** Our bodies are regulated by an internal circadian clock that aligns physiological processes to a 24-hour day-to-night cycle. However, the timing of this rhythm can vary: some people are 'early birds' who prefer mornings, while others are 'night owls' who prefer to wake up and stay up late.

Circadian rhythms have been closely linked to neuropsychiatric conditions like depression, as well as specific psychiatric symptoms such as reduced motivation. Despite this, the circadian clock is seldom considered when investigating the cognitive and motivational changes associated with mental health conditions.

To address this gap, Mehrhof and Nord designed a study to assess motivational differences in the general population and examine whether there were associations between neuropsychiatric symptoms and circadian rhythms. The study focused on effort-based decisions – where individuals choose whether completing a task is worth the effort of the reward – as disruptions in this process often underpin motivational deficits in neuropsychiatric disorders.

Mehrhof and Nord found that individuals with high neuropsychiatric symptoms were less likely to undertake effort-based tasks, consistent with previous studies. Night owls showed the same motivational deficit – even when taking into account neuropsychiatric differences. However, this loss of motivation only occurred when the night owls were tested in the morning. When tested in the evening, their performance matched that of individuals who had an earlier circadian rhythm.

These findings suggest that the circadian clock and neuropsychiatric conditions affect motivation in independent but parallel manners. In addition, testing someone at times of day that misalign with their circadian rhythm may be skewing the results of psychiatric studies. Further research could explore whether aligning treatment schedules and daily routines to a person's internal clock improves motivation and other mental health outcomes.

---

commonly categorized as early, late, or intermediate chronotypes (*Horne and Ostberg, 1976*). A disproportionate number of psychiatric patients have a late chronotype, based on self-report (*Kivelä et al., 2018*) and genetic analysis (*Jones et al., 2019*). Within clinical groups, late chronotype has been linked to depression severity and non-remission (*Chan et al., 2014*), higher rates of psychiatric and general medical comorbidities (*Romo-Nava et al., 2020*), more severe cognitive impairment, and higher symptoms of apathy (*Au and Reece, 2017*; *Coleman and Cain, 2019*). Converging evidence on the importance of circadian alignment in psychiatric pathology has led to proposals of a circadian psychiatric phenotype, either within disorders (*Romo-Nava et al., 2020*; *McGowan and Saunders, 2021*) or cutting across diagnostic categories (*Carpenter et al., 2021*; *Crouse et al., 2021*).

Syndromes of deficient motivational behaviour, such as apathy and anhedonia, are also observed across neuropsychiatric disorders (*Treadway and Zald, 2011*; *Horan et al., 2006*; *Mazza et al., 2009*; *den Brok et al., 2015*), suggesting transdiagnostic relevance (*Husain and Roiser, 2018*). Anhedonia and apathy are associated with worse clinical outcomes (*Ducasse et al., 2018*; *Dimick et al., 2021*) and are poorly targeted by current treatments (*Calabrese et al., 2014*; *Gabbay et al., 2015*; *McMakin et al., 2012*). Empirical work suggests a common underlying neurocognitive mechanism: the integration of costs and benefits during effortful decision-making (*Husain and Roiser, 2018*). Effort-based decision-making is commonly assessed using effort-expenditure tasks: Subjects are asked to decide whether to pursue actions associated with varying levels of effort and reward levels (*Treadway et al., 2009*). Computational models applied to effort-based decision-making tasks provide a formal mathematical estimate of a subject's integration of costs and benefits into a subjective value (*Treadway et al., 2012a*; *Pessiglione et al., 2018*). Higher costs devalue associated rewards, an effect referred to as *effort discounting* (*Sugiwaka and Okouchi, 2004*; *Hartmann et al., 2013*; *Klein-Flügge et al., 2015*; *Białaszek et al., 2017*; *Ostaszewski et al., 2013*). This computational approach enables measurement of inter- and intra-individual differences on distinct aspects of effort-based decision-making.

One key source of individual differences in motivational behaviour and effort-based decision-making is likely dopamine signalling, especially dopaminergic projections from the ventral tegmental area to the ventral striatum (*Husain and Roiser, 2018*). Pre-clinical animal studies show dopamine depletion reduces engagement in effortful behaviour (*Tran et al., 2002*; *Robbins et al., 1983*), while

dopamine enhancement promotes motivational effort exertion (*Cagniard et al., 2006*; *Taylor and Robbins, 1984*). In humans, dopamine depletion reduces willingness to exert effort for reward (*Cawley et al., 2013*; *Venugopalan et al., 2011*), while pharmacological dopamine enhancement increases motivation in effort-based decision-making (*Wardle et al., 2011*; *Soder et al., 2021*). Further, naturally occurring variations in dopamine responsivity are correlated with effort-based decision-making: a higher dopamine responsivity (as quantified with positron emission tomography following *d*-amphetamine administration) is associated with willingness to exert greater effort for larger rewards (*Treadway et al., 2012b*).

Bidirectional links between chronobiology and several neurotransmitter systems have been reported, including dopamine (*Kiehn et al., 2023*). In animals, dopamine transmission and biosynthesis vary diurnally (*Castañeda et al., 2004*; *Ferris et al., 2014*), and growing evidence suggests a bidirectional regulation between dopamine signalling and circadian rhythm (*Weber et al., 2004*; *Hampp et al., 2008*; *Imbesi et al., 2009*). In human studies, dopamine availability, dopamine transporter genes, and dopamine receptors have been linked to proxies of circadian rhythm (*Holst et al., 2014*; *Valomon et al., 2014*; *Zhang et al., 2021*) and circadian-regulating gene polymorphisms (*Shumay et al., 2012*). On a behavioural level, sleep deprivation, poor sleep quality, and insomnia were linked to low motivation in effort-based decision-making (*Boland et al., 2022*; *Boyle et al., 2020*; *Libedinsky et al., 2013*) and evening bright-light exposure enhanced effort willingness, possibly by enhancing dopamine through melatonin suppression (*Bijleveld and Knufinke, 2018*). Early chronotype predicted treatment effect on motivational behaviour in a sample of depressed subjects with comorbid insomnia (*Boland et al., 2019*). Chronotype effects are also reported for other reward decision-making tasks, with late chronotypes showing higher delay discounting (*Evans and Norbury, 2021*), less rational decision-making (*Correa et al., 2020*), and lower willingness to take risks for rewards (*Ingram et al., 2016*; *Hisler et al., 2023*). A circadian effect on decision-making under risk is reported, with the sensitivity to losses decreasing with time-of-day (*Bedder et al., 2023*). This suggests that chronobiology may contribute to individual differences in effort-based decision-making, potentially in parallel ways with neuropsychiatric syndromes.

Here, we tested the relationship between motivational decision-making and three key neuropsychiatric syndromes: anhedonia, apathy, and depression, taking both a transdiagnostic and categorical (diagnostic) approach. To do this, we validate a newly developed effort-expenditure task, designed for online testing, and gamified to increase engagement. Participants completed the effort-expenditure task online, followed by a series of self-report questionnaires.

Next, we pre-registered a follow-up experiment to directly investigate how circadian preference interacts with time-of-day on motivational decision-making, using the same task and computational modelling approach. While this allows us to test how circadian effects on motivational decision-making compare to neuropsychiatric effects, we do not test for possible interactions between neuropsychiatric symptoms and chronobiology. All analyses were pre-registered (except when labelled as exploratory): see https://osf.io/2x3au and https://osf.io/y4fbe.

## Results

### Sample characteristics

Nine hundred and ninety-four participants completed all study components (i.e., demographic questions, effort-expenditure task, self-report questionnaires). After exclusion (see Methods 4.1.5), 958 participants were included in our analyses. We used a stratified recruitment approach to ensure our sample was representative of the UK population in age, sex, and history of psychiatric disorder (*Dercon et al., 2024*; *McManus et al., 2016*; *Office of National Statistics, 2016*); mean questionnaire-based measures were comparable to previous general population studies (*Table 1*).

Questionnaire sum scores highly correlated within groupings of questionnaires targeting psychiatric symptoms, chronobiology, and metabolic health. We also found significant correlations between some, but not all, questionnaires (*Figure 1*).

### Effort-expenditure task

In this novel, online effort-expenditure task (*Figure 2A, B*), subjects were given a series of challenges associated with varying levels of effort and reward. By weighing up efforts against rewards, they decide

**Table 1.** Demographic characteristics and descriptive questionnaire measures in the included sample and excluded participants.

| | Included | Excluded |
|---|---|---|
| Cohort size (%) | 958 (96.4%) | 36 (3.62%) |
| *Demographics* | | |
| Age, mean (SD; range) | 45.00 (15.01; 18–79) | 47.90 (13.60; 20–70) |
| Gender, number (%) | | |
| Male (%) | 470 (49.06) | 12 (33.33) |
| Female (%) | 484 (50.52) | 24 (66.67) |
| Non-binary (%) | 4 (0.42) | 0 (0.0) |
| Ethnicity, number (%) | | |
| White (%) | 852 (88.94) | 28 (77.78) |
| Asian (%) | 53 (5.53) | 4 (11.1) |
| Black (%) | 27 (2.82) | 3 (8.33) |
| Mixed (%) | 18 (1.88) | 1 (2.78) |
| Other (%) | 8 (0.84) | 0 (0.0) |
| SES (/9), median (IQR) | 5 (4–6) | 5 (4–6) |
| *Psychiatric comorbidities* | | |
| Current or past, number (%) | | |
| Any (%) | 264 (27.60) | 5 (13.90) |
| Major depressive disorder (%) | 94 (9.81) | 1 (2.78) |
| Generalized or social anxiety disorder (%) | 195 (20.35) | 2 (5.56) |
| Current antidepressant use, number (%) | 151 (15.80) | 5 (13.9) |
| *Task metrics* | | |
| Testing time, number (%) | | |
| Morning testing (8:00–11:59; %) | 492 (51.40) | 19 (52.80) |
| Evening testing (18:00–21:59; %) | 458 (47.80) | 17 (47.2) |
| Time taken (min), mean (SD; range) | 33.13 (9.63; 22–151) | 37.06 (15.30; 26–105) |
| Mean clicking calibration, mean (SD; range) | 60.6 (16.10; 8–206) | 74.10 (123.00; 0–721) |
| *Psychiatric questionnaire measures* | | |
| SHAPS, mean (SD; range) | 9.15 (6.28; 0–36) | 10.90 (6.97; 1–33) |
| DARS, mean (SD; range) | 54.50 (9.18, 17–68) | 53.70 (9.77, 36–68) |
| AES, mean (SD; range) | 55.70 (9.42; 25–72) | 55.10 (9.51; 37–71) |
| M.I.N.I., current MDD (%) | 56 (5.85) | |
| *Circadian questionnaire measures* | | |
| MEQ, mean (SD; range) | 52.80 (10.6, 18–81) | 52.08 (7.87, 34–71) |
| MCTQ, mean time in min (SD; range) | 03:56 (89 min; 00:14–11:05) | 04:03 (87 min; 01:05–09:05) |
| *Metabolic questionnaire measures* | | |
| BMI, mean (SD; range) | 26.90 (6.29, 15.20–63.30) | 27.02 (5.77, 19.10–46.90) |
| FINDRISC, mean (SD; range) | 7.46 (5.09, 0–25) | 8.56 (5.26, 0–22) |

Note. SES, subjective socioeconomic status; IQR, interquartile range; SHAPS, Snaith Hamilton pleasure scale; DARS, Dimensional Anhedonia Rating Scale; AES, Apathy Evaluation Scale; M.I.N.I., Mini-International Neuropsychiatric Interview; MDD, major depressive disorder; MEQ, Morningness–Eveningness questionnaire; MCTQ, Munich Chronotype Questionnaire; BMI, body mass index; FINDRISC, Finish Diabetes Risk Score.

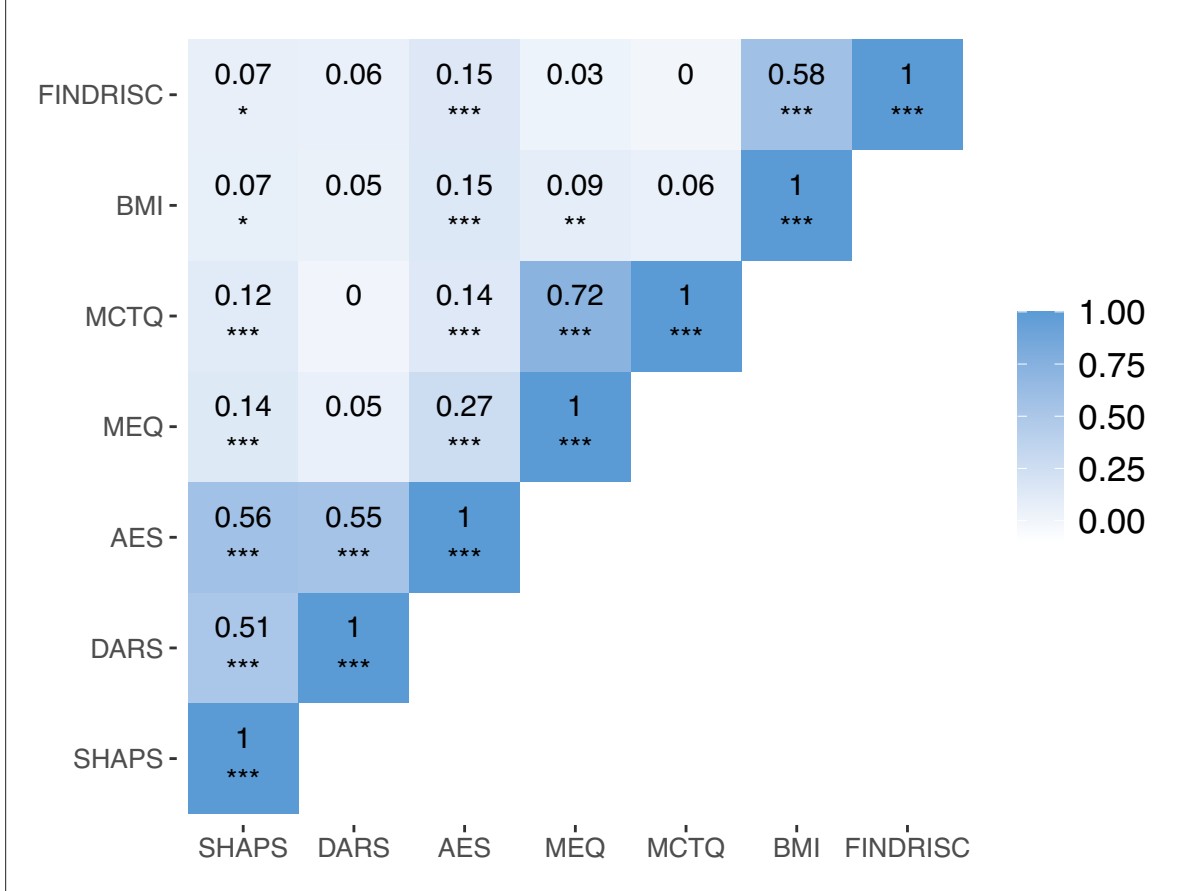

**Figure 1.** Correlations between questionnaire scores. Correlations between questionnaire sum scores for the Snaith Hamilton Pleasure Scale (SHAPS), the Dimensional Anhedonia Rating Scale (DARS), the Apathy Evaluation Scale (AES), Morningness–Eveningness Questionnaire (MEQ), Munich Chronotype Questionnaire (MCTQ), body mass index (BMI), and the Finish Diabetes Risk Score (FINDRISC) (n=958). Asterisks indicate significance: *p < 0.05, **p < 0.01, ***p < 0.001 (not accounting for multiple comparisons). Note that sum scores for the AES and the DARS have been transformed such that increasing scores can be interpreted as higher symptom severity, in line with the SHAPS. Sum scores of the MEQ have been transformed such that higher scores indicate higher eveningness, in line with the MCTQ.

whether to accept or reject challenges. We first use model-agnostic analyses to replicate effects of effort discounting (i.e., devaluation of reward with increasing effort). Next, we took a computational modelling approach to fit economic decision-making models to the task data (*Figure 3A–D*). The models posit efforts and rewards are joined into a subjective value (SV), weighed by individual effort $(\beta_E)$ and reward sensitivity $(\beta_R)$ parameters. The subjective value is then integrated with an individual bias to accept effortful challenges for reward ($\alpha$) parameter to guide decision-making. Specifically, this acceptance bias parameter determines the range at which subjective values are translated to acceptance probabilities: the same subjective value will translate to a higher acceptance probability the higher the acceptance bias.

## Replication of model-agnostic effects

The proportion of accepted trials for each effort–reward combination is plotted in *Figure 2C*. In line with our pre-registered hypotheses, we found significant main effects for effort ($F(1,14367) = 4961.07$, $p < 0.0001$) and reward ($F(1,14367) = 3037.91$, $p < 0.001$), and a significant interaction between the two ($F(1,14367) = 1703.24$, $p < 0.001$). In post hoc ANOVAs, effort effects remained significant at all reward levels (all $p < 0.001$) and reward effects remained significant at all effort levels (all $p < 0.001$). The development of offered effort and reward levels across trials is shown in *Figure 2D*; this shows that as participants generally tend to accept challenges rather than reject them, the implemented staircasing procedure develops towards higher effort and lover reward challenges.

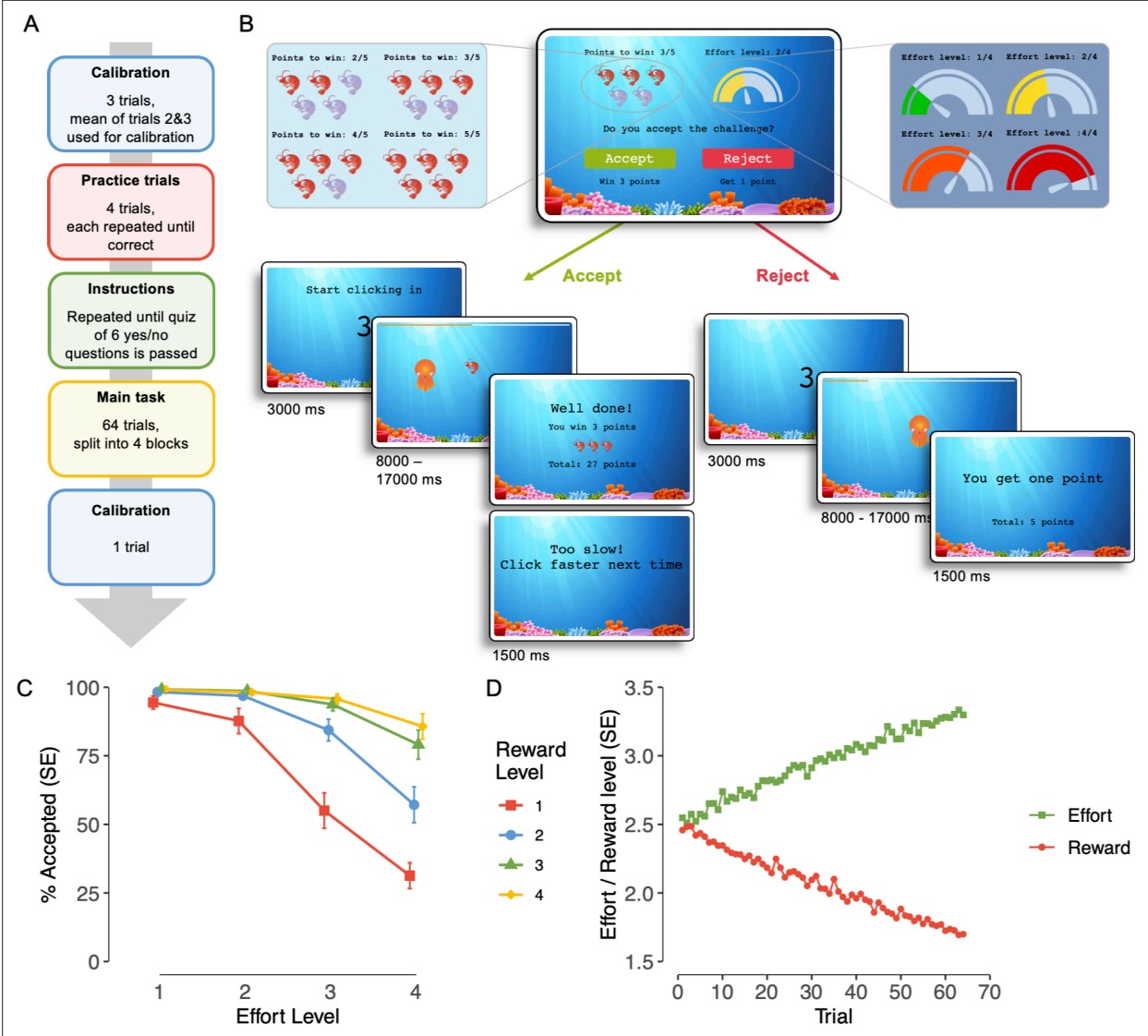

**Figure 2.** Effort-based decision-making: task design and model-agnostic results. (**A**) The task can be divided into four phases: a calibration phase to determine individual clicking capacity to calibrate effort levels, practice trials that participants practice until successful on every effort level, instructions and a quiz that must be passed, and the main task, consisting of 64 trials split into 4 blocks. (**B**) Each trial consists of an offer with a reward (2, 3, 4, or 5 points) and an effort level (1, 2, 3, or 4, scaled to the required clicking speed and time the clicking must be sustained for) that subjects accept or reject. If accepted, a challenge at the respective effort level must be fulfilled for the required time to win the points. If rejected, subjects wait for a matched amount of time and receive one point. (**C**) Proportion of accepted trials, averaged across participants and effort–reward combinations. Error bars indicate standard errors (n = 958). (**D**) Staircasing development of offered effort and reward levels across the task, averaged across participants (n = 958).

The mean success rate of accepted challenges across participants was high (*M* = 98.7%) and varied little between participants (SD = 3.50), indicating feasibility of all effort levels across participants. Comparing clicking calibration results from pre- to post-task, the maximum clicking capacity decreased by 2.34 clicks on average (SD = 14.5). Sixty-two (6.47%) participants reported having deviated from our instructions (i.e., changed the hand and/or finger used to make mouse clicks) throughout the game, but all effects could still be replicated in this subsample: main and interaction effects of effort and reward on the proportion of accepted trials could be replicated in this subsample (all p < 0.001) and there was no significant difference between participants that did or did not report finger switching in the mean percentage of accepted trials (switching: 79.51%, no switching: 76.62%; p = 0.149).

Subjects were engaged with the task, shown by a high rate of challenge acceptance (*M* = 76.80%, SD = 15.20, range = 15.60–100%) and moderate-to-good enjoyment ratings (*M* = 2.56, SD = 0.92;

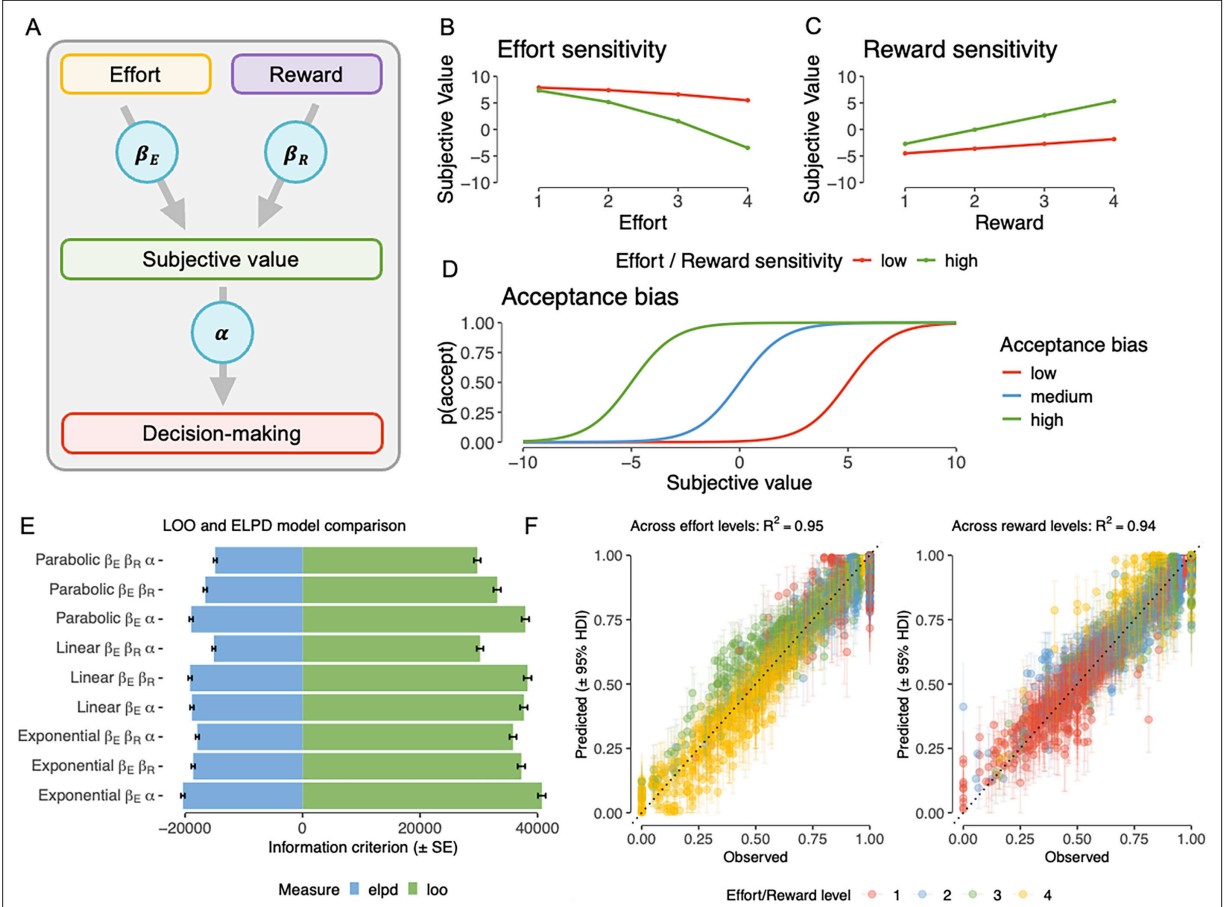

**Figure 3.** Computational modelling: model visualization and model-based results. (**A**) Economic decision-making models posit that efforts and rewards are joined into a subjective value (SV), weighed by individual *effort* ($\beta_E$) and *reward sensitivity* ($\beta_R$) parameters. The SV is then integrated with an acceptance bias parameter and translated to decision-making. (**B, C**) The model suggests that SV decreases as effort increases and increases as reward increases. The magnitude of this relationship depends on the individual effort and reward sensitivity parameters. (**D**) The acceptance bias parameter acts as an intercept to the softmax function, thereby changing the relationship between SV and acceptance probability. (**E**) Model comparison based on leave-out-out information criterion (LOOIC; lower is better) and expected log posterior density (ELPD; higher is better). Error bars indicate standard errors (n = 958). (**F**) Posterior predictive checks for the full parabolic model, comparing observed versus model-predicted subject-wise acceptance proportions across effort levels (left) and reward levels (right). Error bars indicate 95% highest density intervals (n = 958).

on a 0–4 scale). Qualitative data of subjects describing their decision-making process during the task further confirmed high engagement (see Appendix 3).

## Computational modelling

A model space of nine models was considered, varying in the implemented parameter and cost function (see Mathematical definition of the model space for mathematical definitions of all models). Prior to model fitting, parameter recovery confirmed all models yield meaningful parameter estimates (Model validation). All models showed good convergence (effective sample size (ESS) >4223; R-hats <1.002 for all estimates). Model comparison by out-of-sample predictive accuracy identified the model implementing three parameters (acceptance bias $\alpha$, reward sensitivity $\beta_R$, and effort sensitivity $\beta_E$), with a parabolic cost function (subsequently referred to as the *full parabolic model*) as the winning model (leave-one-out information criterion [LOOIC; lower is better] = 29,734.8; expected log posterior density [ELPD; higher is better] = –14,867.4; *Figure 3E*). This was in line with our preregistered hypotheses. Predictive validity of the full parabolic model was validated with posterior predictive checks, showing excellent accordance between observed and model-predicted choice data (across effort levels: $R^2$ = 0.95, across reward levels: $R^2$ = 0.94; *Figure 3F*).

### Test–retest reliability

We validated the task in a smaller in-person sample (*N* = 30, tested twice ~7 days apart, holding time-of-day at testing constant) to assess test–retest reliability of parameter estimates, showing moderate to excellent reliability for all parameters (i.e., all intraclass correlation coefficients > 0.4, all p < 0.01). Parameter estimates from modelling the data at one session predicted subjects' choices at the other session better than chance and better than group-level parameters predictions (all p < 0.01) (*Mkrtchian et al., 2023*) (full details reported in Appendix 2).

## Transdiagnostic analysis: questionnaire measures predict effort-based decision-making

We used partial least squares (PLS) regression to relate individual-level mean posterior parameter values resulting from the model fitting of the full parabolic model to the questionnaire measures. To explore individual effects post hoc, we followed up on effects found in the PLS regression using Bayesian generalized linear models (GLMs), controlling for age and gender.

### Acceptance bias

The acceptance bias was best predicted by a model with one component, with its highest factor loadings from psychiatric measures increasing values indicate symptom severity; SHAPS *Snaith et al., 1995*: –0.665; Apathy Evaluation Scale [AES] *Marin et al., 1991*: –0.588; Dimensional Anhedonia Rating Scale [DARS] *Rizvi et al., 2015*: –0.487. Weaker loadings were found for circadian measures higher values indicate later chronotype; Morningness–Eveningness Questionnaire [MEQ] *Horne and*

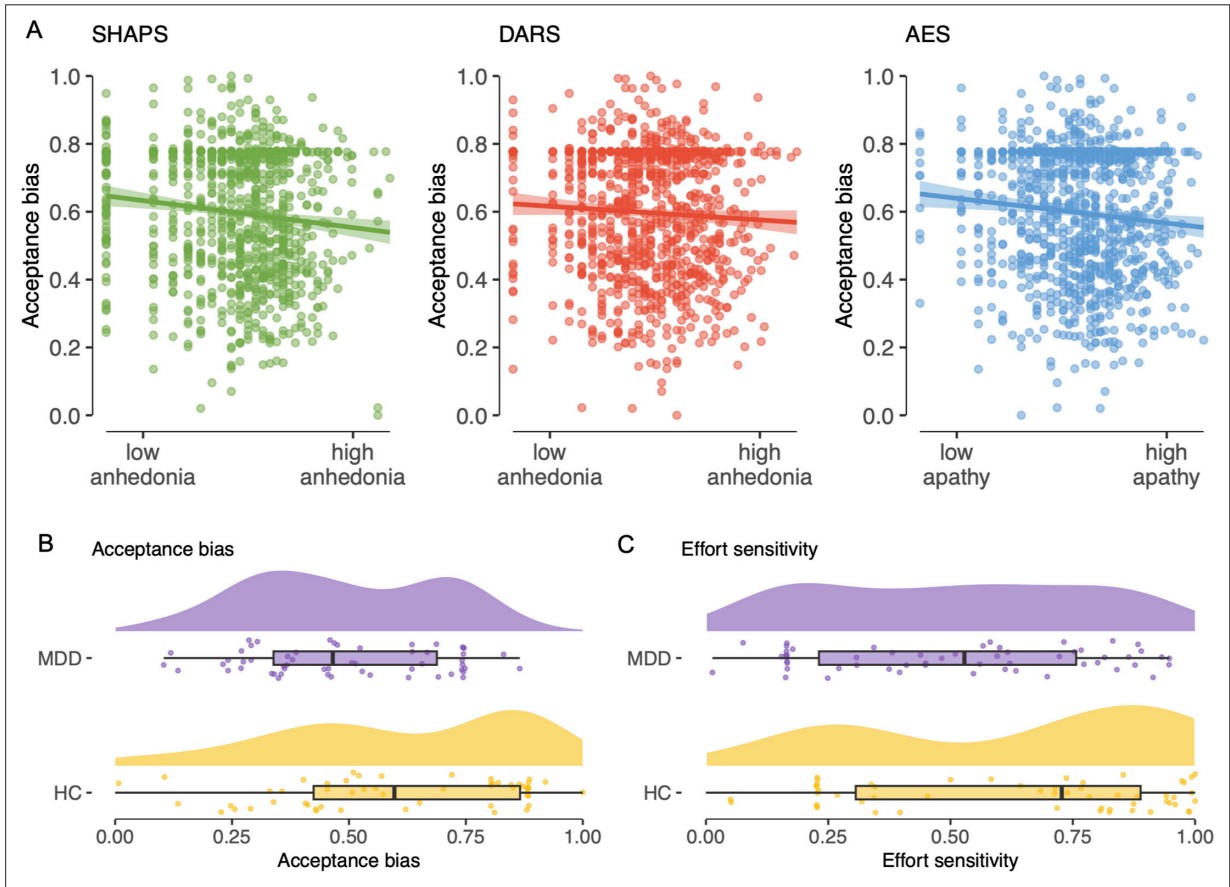

**Figure 4.** Associations between task parameter estimates and psychiatric measures. (**A**) Visualizations of associations between the acceptance bias task parameter and the Snaith–Hamilton Pleasure Scale (SHAPS), the Dimensional Anhedonia Rating Scale (DARS) (*Rizvi et al., 2015*), and the Apathy Evaluation Scale (AES) (*Marin et al., 1991*). (**B, C**) Comparison of acceptance bias (left) and effort sensitivity (right) between a sample of participants meeting criteria for current major depressive disorder (MDD; purple, upper) on the the Mini-International Neuropsychiatric Interview 7.0.1 (M.I.N.I) (*Lecrubier et al., 1997*) and age- and gender-matched controls (yellow, lower).

*Ostberg, 1976*: –0.262; Munich Chronotype Questionnaire [MCTQ] *Roenneberg et al., 2003*: –0.117 and metabolic measures higher values indicate higher metabolic risk; body mass index [BMI]: –0.115; Finnish Type-2 Diabetes Risk Score questionnaire [FINDRISC] *Lindström and Tuomilehto, 2003*: –0.068. Permutation testing indicated the predictive value of the resulting component (with factor loadings described above) was significant out-of-sample (root-mean-squared error [RMSE] = 0.203, p = 0.001).

Bayesian GLMs confirmed evidence for psychiatric questionnaire measures predicting acceptance bias (SHAPS: $M = -0.109$; 95% highest density interval (HDI) = [–0.17,–0.04]; AES: $M = -0.096$; 95% HDI = [–0.15,–0.03]; DARS: $M = -0.061$; 95% HDI = [–0.13,–0.01]; *Figure 4A*). Post hoc GLMs on DARS subscales showed an effect for the sensory subscale ($M = -0.050$; 95% HDI = [–0.10,–0.01]). This result of neuropsychiatric symptoms predicting a lower acceptance bias is in line with our pre-registered hypothesis. For the MEQ (95% HDI = [–0.09,0.06]), MCTQ (95% HDI = [–0.17,0.05]), BMI (95% HDI = [–0.19,0.01]), and FINDRISC (95% HDI = [–0.09,0.03]) no relationship with acceptance bias was found, consistent with the smaller magnitude of reported component loadings from the PLS regression. This null finding for dimensional measures of circadian rhythm and metabolic health was not in line with our pre-registered hypotheses.

### Effort sensitivity
For effort sensitivity, the intercept-only model outperformed models incorporating questionnaire predictors based on RMSE.

### Reward sensitivity
For reward sensitivity, the intercept-only model outperformed models incorporating questionnaire predictors based on RMSE. This result was not in line with our pre-registered expectations.

### Questionnaire measures predict model-agnostic task measures
Both SHAPS ($M = -0.07$; 95% HDI = [–0.12,–0.03]) and AES ($M = -0.05$; 95% HDI = [–0.10,–0.002]) sum scores could predict the proportion of accepted trials averaged across effort and reward levels (*Appendix 3—figure 1*).

## Diagnostic analysis: depressed and healthy subjects differ in effort-based decision-making

In an exploratory analysis, we compared a sample of $N = 56$ participants that met criteria for current major depressive disorder (MDD), to 56 healthy controls (HC), matched by age (MDD: $M = 37.07$; HC: $M = 37.09$, p = 0.99) and gender (MDD: 31 female, 23 male, 2 non-binary; HC: 32 female, 22 male, 2 non-binary; p = 0.98). Effort-discounting effects were confirmed in both groups. For both groups, model fitting and comparison identified the full parabolic model as the best-fitting model. We used age- and gender-controlled Bayesian GLMs to compare individual-level mean posterior parameter values between groups.

### Acceptance bias
As in our transdiagnostic analyses of continuous neuropsychiatric measures (see Transdiagnostic analysis: questionnaire measures predict effort-based decision-making), we found evidence for a lower acceptance bias parameter in the MDD group compared to HCs ($M = -0.111$, 95% HDI=[–0.20,–0.03]) (*Figure 4B*). This result confirmed our pre-registered hypothesis.

### Effort sensitivity
Unlike our transdiagnostic analyses, we also found evidence for lower effort sensitivity in the MDD group compared to HCs ($M = -0.111$, 95% HDI = [–0.22,–0.01]) (*Figure 4C*).

### Reward sensitivity
There was no evidence for a group difference in reward sensitivity (95% HDI = [–0.07,0.11]), as in our transdiagnostic analyses.

## Circadian measures affect effort-based decision-making

Due to our hypothesized interaction between circadian preference and time-of-day, testing was conducted in two specified time windows: morning (08:00–11:59) and evening (18:00–21:59), resulting in a binary time-of-day measure (morning vs. evening testing). A total of 492 participants completed the study in the morning testing window and 458 in the evening testing window. We used the two chronotype questionnaires to identify two established circadian phenotypes: 'early' or 'late' chronotype (see Investigating circadian effects), behavioural categories indicating underlying chronobiological differences (*Jones et al., 2019*; *Horne and Ostberg, 1976*; *Roenneberg et al., 2003*). These classifications result in four subsample groups, with 89 early chronotypes (morning testing: *n* = 63; evening testing: *n* = 26) and 75 late chronotypes (morning testing: *n* = 20; evening testing: *n* = 55).

Bayesian GLMs, controlling for age and gender, predicting task parameters by time-of-day and chronotype showed effects of chronotype on reward sensitivity (i.e., those with a late chronotype had a higher reward sensitivity; $M = 0.325$, 95% HDI = [0.19,0.46]) and acceptance bias (higher acceptance bias in early chronotypes; $M = -0.248$, 95% HDI = [–0.37,–0.11]), as well as an interaction between chronotype and time-of-day on acceptance bias ($M = 0.309$, 95% HDI = [0.15,0.48]).

### Additional pre-registered data collection

As these analyses rely on unevenly distributed subsamples, we conducted an additional, pre-registered data collection to replicate and extend these findings (https://osf.io/y4fbe). We screened participants

**Table 2.** Demographic characteristics and descriptive questionnaire measures in the early and late chronotype participants.

|  | Early chronotype | Late chronotype | Significance |
|---|---|---|---|
| Sample size (%) | 102 (51.78%) | 95 (48.22%) |  |
| *Demographics* |  |  |  |
| Age, mean (SD; range) | 51.80 (14.10; 20–78) | 35.80 (14.40; 19–68) | p < 0.001 |
| Gender, number (%) |  |  | p < 0.05 |
| Male | 42 (41.18) | 55 (57.89) |  |
| Female | 60 (58.82) | 40 (42.11) |  |
| *Testing time* |  |  |  |
| Start testing time, number (%) |  |  | p < 0.01 |
| Morning testing (8:00–11:59) | 63 (31.98) | 38 (19.29) |  |
| Evening testing (18:00–21:59) | 39 (19.80) | 57 (28.93) |  |
| *Psychiatric comorbidities* |  |  |  |
| Current or past, number (%) |  |  |  |
| Any | 22 (21.60) | 40 (42.10) | p < 0.01 |
| Major depressive disorder | 4 (3.92) | 22 (23.16) | p < 0.001 |
| Generalized or social anxiety disorder | 18 (17.65) | 24 (25.26) | p = 0.258 |
| Current antidepressant use, number (%) | 9 (8.82) | 26 (27.40) | p < 0.1 |
| *Psychiatric questionnaire measures* |  |  |  |
| SHAPS, mean (SD; range) | 9.65 (6.38) | 11.80 (5.92) | p < 0.05 |
| DARS, mean (SD; range) | 54.00 (9.37) | 52.70 (9.39) | p = 0.322 |
| AES, mean (SD; range) | 56.00 (9.72) | 50.60 (10.10) | p < 0.001 |
| M.I.N.I., current MDD (%) | 3 (2.94) | 15 (15.79) | p < 0.01 |

Note. SES, subjective socioeconomic status; IQR, interquartile range; SHAPS, Snaith–Hamilton Pleasure Scale; DARS, Dimensional Anhedonia Rating Scale; AES, Apathy Evaluation Scale; M.I.N.I., Mini-International Neuropsychiatric Interview; MDD, major depressive disorder; MEQ, Morningness–Eveningness Questionnaire; MCTQ, Munich Chronotype Questionnaire; BMI, body mass index; FINDRISC, Finish Diabeted Risc Score.

for their chronotype and then invited early chronotypes to take part in our study in the evening testing window, and late chronotypes in the morning testing window (see Additional data collection).

Using our pre-registered Bayesian stopping rule, we tested 13 early chronotype participants and 20 late chronotype participants. The data were then combined with the data from our main data collection, resulting in a full sample of $n = 197$ participants that was used for subsequent chronotype analyses (see *Table 2* for sample characteristics and statistical significance of differences).

### Acceptance bias

Late chronotypes showed a lower acceptance bias than early chronotypes ($M = −0.11$, 95% HDI = [−0.22,−0.02])—comparable to effects of transdiagnostic measures of apathy and anhedonia, as well as diagnostic criteria for depression. Crucially, we found acceptance bias was modulated by an interaction between chronotype and time-of-day ($M = 0.19$, 95% HDI = [0.05,0.33]): post hoc GLMs in each chronotype group showed this was driven by a time-of-day effect within late, rather than early, chronotype participants ($M = 0.12$, 95% HDI = [0.02,0.22], such that late chronotype participants showed

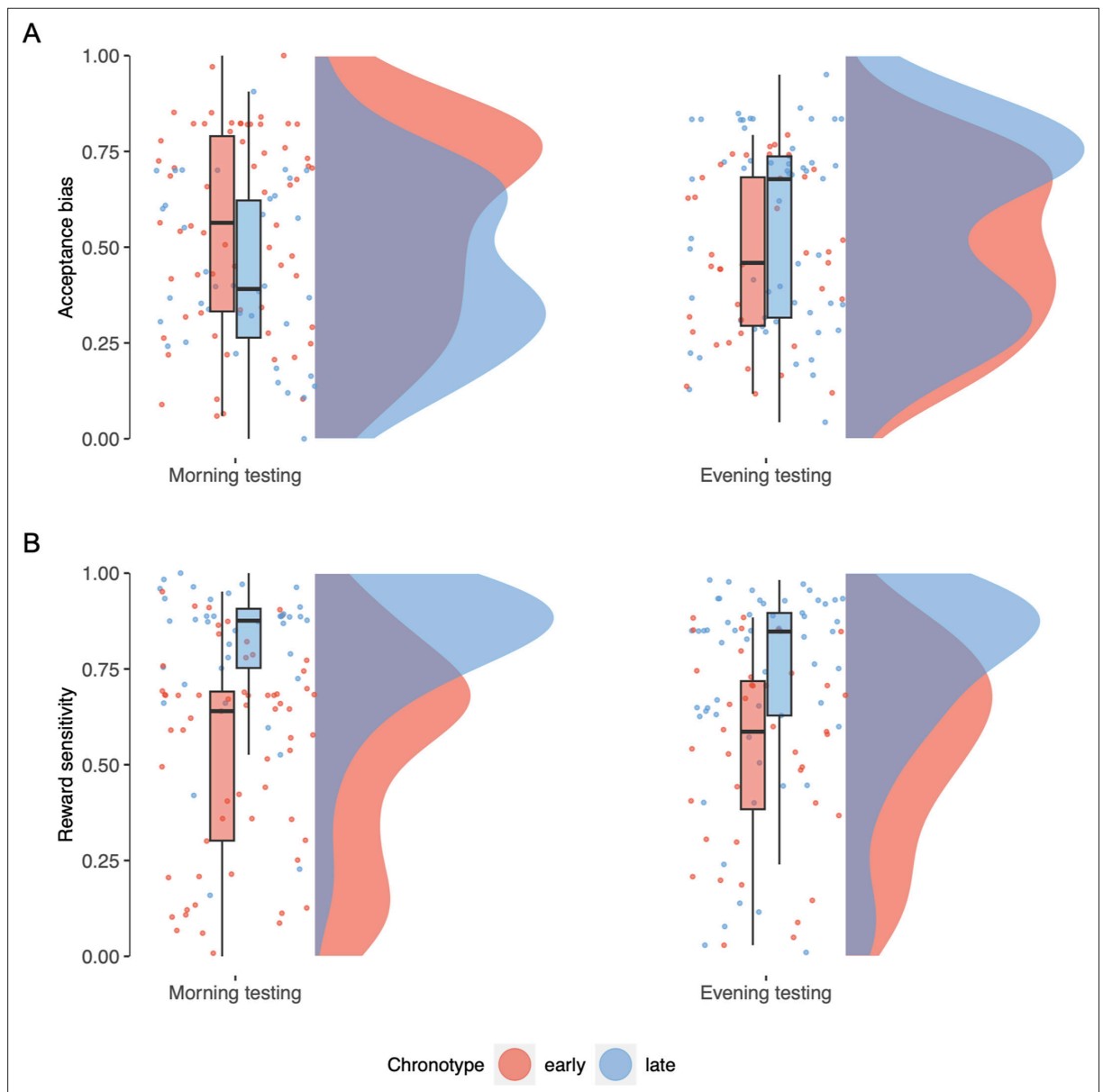

**Figure 5.** Effects of chronotype and time-of-day on task parameter estimates. (**A**) Effect of chronotype and time-of-day on reward sensitivity parameter estimates. (**B**) Effect of chronotype and time-of-day on acceptance bias parameter estimates.

a lower acceptance bias in the morning testing sessions, and a higher acceptance bias in the evening testing sessions; early chronotype: 95% HDI = [–0.16,0.04]) (*Figure 5A*). These results of a main effect and an interaction effect of chronotype on acceptance bias confirmed our pre-registered hypothesis.

## Neuropsychiatric symptoms and circadian measures have separable effects on acceptance bias

### Acceptance bias

Exploratory analyses testing for the effects of neuropsychiatric questionnaires on acceptance bias in the subsamples of early and late chronotypes confirmed the predictive value of the SHAPS ($M$ = −0.24, 95% HDI = [–0.42,–0.06]), the DARS ($M$ = −0.16, 95% HDI = [–0.31,–0.01]), and the AES ($M$ = −0.18, 95% HDI = [–0.32,–0.02]) on acceptance bias.

For the SHAPS, we find that when adding the measures of chronotype and time-of-day back into the GLMs, the main effect of the SHAPS ($M$ = −0.26, 95% HDI = [–0.43,–0.07]), the main effect of chronotype ($M$ = −0.11, 95% HDI = [–0.22,–0.01]), and the interaction effect of chronotype and time-of-day ($M$ = 0.20, 95% HDI = [0.07,0.34]) on acceptance bias remain. Model comparison by LOOIC reveals acceptance bias is best predicted by the model including the SHAPS, chronotype and time-of-day as predictors, followed by the model including only the SHAPS. Note that this approach to model comparison penalizes models for increasing complexity.

Repeating these steps with the DARS, the main effect of the DARS is found numerically, but the 95% HDI just includes 0 ($M$ = −0.15, 95% HDI = [–0.30, 0.002]). The main effect of chronotype ($M$ = −0.11, 95% HDI = [–0.21,–0.01]), and the interaction effect of chronotype and time-of-day ($M$ = 0.18, 95% HDI = [0.05,0.33]) on acceptance bias remain. Model comparison identifies the model including the DARS and circadian measures as the best model, followed by the model including only the DARS.

For the AES, the main effect of the AES is found ($M$ = −0.19, 95% HDI = [–0.35,–0.04]). For the main effect of chronotype, the 95% narrowly includes 0 ($M$ = −0.10, 95% HDI = [–0.21, 0.002]), while the interaction effect of chronotype and time-of-day ($M$ = 0.20, 95% HDI = [0.07,0.34]) on acceptance bias remains. Model comparison identifies the model including the AES and circadian measures as the best model, followed by the model including only the AES.

### Effort sensitivity

We found no evidence for circadian or time-of-day effects on effort sensitivity (chronotype main effect: 95% HDI = [–0.06,0.18], time-of-day main effect: 95% HDI = [–0.08,0.13]).

### Reward sensitivity

Participants with an early chronotype had a lower reward sensitivity parameter than those with a late chronotype ($M$ = 0.27, 95% HDI = [0.16,0.38]). We found no effect of time-of-day on reward sensitivity (95% HDI = [–0.09,0.11]) (*Figure 5B*). These results were in line with our pre-registered hypotheses .

## Discussion

Various neuropsychiatric disorders are marked by disruptions in circadian rhythm, such as a late chronotype. However, research has rarely investigated how transdiagnostic mechanisms underlying neuropsychiatric conditions may relate to inter-individual differences in circadian rhythm. Here, combining a large-scale online study with computational modelling, we replicate and extend previous work linking anhedonia, apathy, and depression to a lower bias to accept effort for reward. Crucially, we found participants with a late compared to early chronotype show the same decrease in acceptance bias. Moreover, by testing participants at chronotype-compatible and -incompatible times of day, we discovered late chronotypes show a decreased acceptance bias to exert effort for reward when tested in the morning compared to evening. This reveals neuropsychiatric symptoms and chronotype (interacting with time-of-testing) show paralleling effects on effort-based decision-making. Our results demonstrate a crucial role for circadian rhythm in computational psychiatry, potentially affecting our assessment and treatment of neurocognitive mechanisms.

We replicate and extend effects of aberrant effort-based decision-making in neuropsychiatric syndromes in a large, broadly population-representative sample. Our finding that dimensional measures of apathy and anhedonia predict acceptance bias, a computational parameter describing

someone's tendency to exert effort for reward, aligns with previous reports of impaired effort-based decision-making in psychiatric (*Treadway et al., 2012a*; *Barch et al., 2014*; *Berwian et al., 2020*; *Cléry-Melin et al., 2011*; *Fervaha et al., 2013*; *Gold et al., 2013*; *Hershenberg et al., 2016*; *Wolf et al., 2014*; *Yang et al., 2014*; *Zou et al., 2020*) and neurodegenerative populations (*Chong et al., 2015*; *Chong et al., 2018*; *Le Bouc et al., 2016*; *Le Bouc et al., 2023*; *Le Heron et al., 2018*), as well as studies linking effort-based decision-making with apathy and anhedonia specifically. The positive link between effort-based decision-making and apathy and anhedonia has been observed in both patients (*Barch et al., 2014*; *Yang et al., 2014*; *Tran et al., 2021*) and HCs (*Barch et al., 2014*; *Bonnelle et al., 2015*; *Jurgelis et al., 2021*), though some did not find this effect (*Zou et al., 2020*).

Our work supports previous theories that impaired effort-based decision-making represents a common, transdiagnostic mechanism across the psychiatric and neurological syndromes of anhedonia and apathy (respectively). We found corresponding effects of apathy and anhedonia on the same computational parameter—acceptance bias—reinforcing the suggestion of possible shared mechanistic underpinnings of the two motivational syndromes (*Husain and Roiser, 2018*). Aberrant effort-based decision-making may manifest behaviourally as deficient motivation, a symptom category that cuts across traditional disease boundaries of psychiatric, neurological, and neurodevelopmental disorders (*Pessiglione et al., 2018*).

Our categorical (diagnostic-criteria based) analysis comparing depressed to healthy subjects likewise found depressed patients showed a lower acceptance bias, echoing our dimensional results in apathy and anhedonia. In addition, our categorical analysis revealed a distinct effect of group on effort sensitivity: depressed subjects had lower effort sensitivity, meaning their decisions were less influenced by effort changes. Possibly, this effect stems from decreased perceived differences in effort levels, as recently reported (*Silvia et al., 2016*), indicating there are both dimensional (transdiagnostic) and potentially some diagnosis-specific effects of mental health on effort-based decision-making.

It is possible that a higher acceptance bias reflects a more optimistic assessment of future task success, in line with work on the optimism bias (*Korn et al., 2014*); however, our task intentionally minimized unsuccessful trials by titrating effort and reward; future studies should explore this more directly.

We also found circadian effects on effort-based decision-making, paralleling those of apathy, anhedonia, and depression measures in both the affected neurocomputational parameter and the direction of effect. We observed a difference in acceptance bias between chronotypes, with late chronotypes showing a lower tendency to accept to exert effort for reward. Previous studies have suggested late chronotypes were also less accepting of delays (*Evans and Norbury, 2021*) and risk for reward (*Ingram et al., 2016*; *Hisler et al., 2023*).

Most importantly, we found an interaction between chronotype and time-of-day in a synchrony effect manner: early and late chronotypes showed a *higher* acceptance bias towards accepting effort for reward *at their preferred time-of-day*. This effect was driven by the late chronotype group, who showed a markedly lower acceptance bias in the morning, but a higher tendency in the evening. This suggests that chronotype effects on neurocomputational parameters such as acceptance bias depend on time-of-testing. Synchrony effects have previously been observed in other cognitive domains including inhibitory control (*May and Hasher, 1998*), attention (*Goldstein et al., 2007*), learning (*Lehmann et al., 2013*), and memory (*Barner et al., 2019*). One interpretation of our cognitive synchrony effects may be that late chronotype participants show a diminished ability to adapt to suboptimal time-of-day due to reduced cognitive resources (*Nowack and Van Der Meer, 2018*).

We also report a distinct effect of chronotype on effort-based decision-making that is not paralleled by effects of neuropsychiatric symptoms, nor dependent on time-of-day. Compared to early chronotypes, late chronotypes were more guided by differences in reward value, indicated by higher reward sensitivity parameters. Previous studies report altered reward functioning in late chronotypes, who show a reduced reactivity to reward in the medial prefrontal cortex, a key component of reward circuitry (*Forbes et al., 2012*; *Hasler et al., 2013*; *Holm et al., 2009*). Note this is not incompatible with higher reward sensitivity due to our modelling approach, in which higher reward sensitivity does not imply higher reward valuation, but rather larger subjective value differences between reward levels. Therefore, reduced reactivity to reward could be compatible with late chronotypes devaluing low reward levels more, which in our models would emerge as a reduced reward sensitivity parameter.

It is striking that the effects of neuropsychiatric symptoms on effort-based decision-making largely are paralleled by circadian effects on the same neurocomputational parameter. Exploratory analyses predicting acceptance bias by neuropsychiatric symptoms and circadian measures simultaneously indicate the effects go beyond recapitulating each other, but rather explain separable parts of the variance in acceptance bias. Overall, our results raise the possibility of altered effort–reward processing as a critical mechanism linking neuropsychiatric conditions and circadian rhythm. Previous research demonstrated depressed patients with an evening chronotype show increased diurnal mood variation (*Chen et al., 2022*). Our finding of time-of-day differences in acceptance bias among late chronotypes illustrates a potential cognitive underpinning for the observed diurnal characteristic within depression and late chronotype. Together, these findings support the idea of a circadian psychiatric phenotype (*Romo-Nava et al., 2020*; *McGowan and Saunders, 2021*), which should be considered in measurement (e.g., design of computational psychiatry studies) and potentially treatment (e.g., administration of motivation-based psychological interventions, which could be timed compatibly with chronotype).

To our surprise, we did not find statistical evidence for a relationship between effort-based decision-making and measures of metabolic health (BMI and risk for type-2 diabetes). Our analyses linking BMI to acceptance bias reveal a numeric effect in line with our hypothesis: a higher BMI relating to a lower acceptance bias. However, the 95% HDI for this effect narrowly included zero (95% HDI = [–0.19,0.01]). Possibly, our sample did not have sufficient variance in metabolic health to detect dimensional metabolic effects in a current general population sample. A recent study by our group investigates the same neurocomputational parameters of effort-based decision-making in participants with type-2 diabetes and non-diabetic controls matched by age, gender, and physical activity (*Mehrhof et al., 2024*). We report a group effect on the acceptance bias parameter, with type-2 diabetic patients showing a lower tendency to exert effort for reward.

Our study results should be considered in light of a few limitations. First, we used online self-report measures of neuropsychiatric symptoms and depression status. There has been a large shift towards online data collection in psychiatric research, and while online data are undoubtedly noisier, results (including our own, presented in the Appendix) usually show excellent accordance with lab-based studies (*Gillan and Daw, 2016*). Similarly, we lack biological measures of circadian rhythm, the gold standard of chronotype assessment. However, this concern might be mitigated by previous reports of high covariance between biological- and questionnaire-based circadian measures (*Kantermann et al., 2015*; *Santisteban et al., 2018*), as well as significant chronobiological differences between the questionnaire-determined chronotypes (*Bailey and Heitkemper, 2001*; *Nebel et al., 1996*) we use in our key findings. Nevertheless, future work should incorporate biological measures in attempts to replicate circadian effects on effort-based decision-making. This could take the form of identifying chronotypes by DNA analysis or dim-light melatonin onset, or continuous measurements of circadian proxies, such as core body temperature, heart rate, or actigraphy.

Note also that our time-of-day effects are limited by a between-subjects study design (i.e., the same participants were not tested in morning and evening sessions). It will be interesting to explore such diurnal variation in effort-based decision-making within individuals. The newly developed effort-expenditure task we present here may lend itself particularly well to such endeavours. First, it allows remote testing, meaning subjects can complete the task at different times of the day without in-person testing. Second, we demonstrated good test–retest reliability of task measures when time-of-testing was held constant within participants. This good test–retest reliability of our task contrasts with recent reports of poor test–retest reliability of other tasks and computational modelling parameters (*Vrizzi et al., 2023*).

Our reported analyses investigating neuropsychiatric and circadian effects on effort-based decision-making simultaneously are exploratory, as our study design was not ideally set out to examine this. Further work is needed to disentangle *separable* effects of neuropsychiatric and circadian measures on effort-based decision-making. One approach could be a group-based study design enabling the dissociation of the two effects (e.g., examining high-anhedonia participants with early chronotypes and low-anhedonia patients with late chronotypes, as well as the respective other, more common groupings, and testing each group in the morning and evening to examine time-of-day interactions with both anhedonia and chronotype).

Finally, we would like to note that as our study is based on a general population sample, rather than a clinical one. Hence, we cannot speak to transdiagnosticity on the level of multiple *diagnostic categories.*

Taken together, our results implicate circadian rhythm as an important factor in effort-based decision-making and its relationship to neuropsychiatric conditions. These results have implications for research, clinical interventions, and policy. We demonstrate that neuropsychiatric effects on effort-based decision-making are paralleled by effects of circadian rhythm and time-of-day. Exploratory analyses suggest these effects account for separable parts of the variance in effort-based decision-making. It unlikely that effects of neuropsychiatric effects on effort-based decision-making reported here and in previous literature are a spurious result due to multicollinearity with chronotype. Yet, not accounting for chronotype and time-of-testing, which is the predominant practice in the field, could affect results. This could take the form of either inflating or depressing results in the existing literature. On the one hand, reported neuropsychiatric effects may be inflated by systematic circadian differences between participants (i.e., overrepresentation of late chronotype in patient samples), which could be further amplified by time-of-testing (often the morning, incompatible with late chronotypes, and producing motivational impairments on neurocognitive measures). On the other hand, true effects may be masked by interactions between chronotype and time-of-day: Testing psychiatric subjects with a late chronotype in the evening (e.g., as a consequence of subject-selected testing times) may paint a false picture of group equivalence, as researchers are only observing part of a daily trajectory.

Our growing understanding of the relationship between circadian rhythm and neuropsychiatry may allow for critical advances in improving therapeutic outcomes from treatments (*Bhatnagar et al., 2023*; *Meuret et al., 2016*). Such advances are particularly called for in the case of symptoms of apathy and anhedonia, as current treatments often fail to improve motivational deficits (*Calabrese et al., 2014*; *Gabbay et al., 2015*; *McMakin et al., 2012*), but could potentially be coupled with a patient's chronotype to increase efficacy. At minimum, clinical trials predicting change in motivational measures, such as effort-based decision-making, should assess patients at similar times of day, as this could reduce or inflate treatment effects.

Circadian rhythm and neuropsychiatric syndromes may affect motivation via parallel, as well as distinct, mechanisms—but crucially, this overlap is dependent on time-of-testing. Our work suggests that chronotype and time-of-testing are essential variables to consider in future effort-based decision-making experiments, particularly those measuring effort-based decision-making in patient groups, such as those with depression, high apathy, or high anhedonia. Beyond experimental work, future interventions should consider the role of chronotype in measurement and modulation of motivation.

## Materials and methods
### Study protocol
After providing demographics and basic medical history, subjects completed an effort-expenditure task, followed by a battery of self-report questionnaires. The study was coded in JavaScript, using Phaser v.3.50.0 for the task and jsPsych (*de Leeuw, 2015*) for questionnaires. All experimental materials are publicly available at https://doi.org/10.5281/zenodo.15068968.

### Recruitment
We recruited participants using Prolific (*Palan and Schitter, 2018*), in September 2022. Data were collected on weekdays, in specified daily time windows (*morning testing*: 08:00–11:59; *evening testing*: 18:00–21:59). To sample participants broadly representative of the UK population in age, sex, and history of psychiatric disorder, we implemented a previously described procedure (*Dercon et al., 2024*) using Prolific pre-screeners to obtain batches of participants aimed to match target numbers calculated based on UK population data.

Nine hundred and ninety-four participants completed all components and were paid a fixed rate of £6. A bonus of £10 was paid to ten participants. Subjects were told they could increase their chances of winning the bonus by engaging well with the study (e.g., reading questions carefully, following task instructions).

## Effort-expenditure task

We developed a new effort-expenditure task that allowed us to assess effort-based decision-making in a remote setting; this task was also tested in-person to assess test–retest reliability. To increase engagement, we gamified the task to take place in an underwater setting and each challenge is framed as a race in which an octopus catches a shrimp. The task structure is shown in *Figure 2A* and the trial-level structure in *Figure 2B*.

The task began with an individual calibration phase to standardize maximum effort capacity, followed by the main task, which used a semi-adaptive staircase design to maximize the informative value of each choice.

For the calibration, subjects were prompted to collect points by clicking as fast as possible for 10 s, repeated three times. The second and third repetitions were then averaged to serve as the maximum clicking capacity reference for the main task. One calibration trial was repeated at the end of the main task to monitor any notable changes in clicking capacity. Then, subjects were familiarized with their individually calibrated effort levels during a practice phase of the task. Effort levels were scaled to a given participant's mean clicking speed (based on the calibration phase), and the time clicking must be sustained for. We used four effort levels, corresponding to a clicking speed at 30% of a participant's maximal capacity for 8 s (level 1), 50% for 11 s (level 2), 70% for 14 s (level 3), and 90% for 17 s (level 4). Therefore, in each trial, participants had to fulfil a certain number of mouse clicks (dependent on their capacity and the effort level) in a specific time (dependent on the effort level). Subjects were instructed to make mouse clicks with the finger they normally use, and to not change fingers throughout the task (compliance was checked at the end of the main task). In the practice phase, all effort levels were completed without reward associations, and failed levels were repeated until subjects succeed at each level. If a subject failed a level twice, the clicking capacity reference was adjusted to the speed reached in the practice. Finally, subjects needed to pass a six-question quiz to ensure task instructions were fully understood. If a subject failed any question on the quiz, they were returned to the instruction screens and re-took the quiz until all questions were answered correctly.

The main task took a binary-choice design: In each trial, participants accepted or rejected a challenge associated with one of the four specific effort levels and rewards. Reward was conceptualized as points (shrimp caught by the octopus) that could be collected in that trial. The points to win per challenge varied between four levels (2, 3, 4, or 5 points). If a subject accepted a given challenge, they needed to achieve the given effort level to win the associated points. If a subject rejected a given challenge, they waited and received one point, with waiting times matched to the respective effort level to prevent confounding with delay discounting. Participants were able to infer their clicking progress from the distance between the octopus and the shrimp and the remaining time was indicated by a time bar.

Subjects completed 64 trials, split into four blocks of 16 trials. For each subject, trial-by-trial presentation of effort–reward combinations were made semi-adaptively by 16 randomly interleaved staircases. Each of the 16 possible offers (4 effort levels × 4 reward levels) served as the starting point of one of the 16 staircase. Within each staircase, after a subject accepted a challenge, the next trial's offer on that staircase was adjusted (by increasing effort or decreasing reward). After a subject rejected a challenge, the next offer on that staircase was adjusted by decreasing effort or increasing reward. This ensured subjects received each effort–reward combination at least once (as each participant completed all 16 staircases), while individualizing trial presentation to maximize the trials' informative value. Therefore, in practice, even in the case of a subject rejecing all offers (and hence the staircasing procedures always adapting by decreasing effort or increasing reward), the full range of effort–reward combinations will be represented in the task across the startingpoints of all staircases (and therefore before adaption takeplace).

## Self-report questionnaires

Subjects completed a questionnaire battery assessing mental and physical health, presented in a randomized order. We assessed anhedonia using the SHAPS (*Snaith et al., 1995*), as well as the DARS (*Rizvi et al., 2015*). Apathy was assessed with the AES (*Marin et al., 1991*). Additionally, we screened participants for meeting diagnostic criteria for current, past, or recurrent MDD using the Mini-International Neuropsychiatric Interview (M.I.N.I.) (*Lecrubier et al., 1997*). Two questionnaires targeted circadian rhythm: the MEQ (*Horne and Ostberg, 1976*) and the MCTQ (*Rizvi et al., 2015*).

Metabolic health was assessed by collecting self-reported height and weight, used to calculate BMI. Additionally, the FINDRISC (*Lindström and Tuomilehto, 2003*) was used to calculate individual risk scores for metabolic disease. Finally, the International Physical Activity Questionnaire (IPAQ) (*Craig et al., 2003*) was included for exploratory investigations of physical activity.

### Compliance checks and exclusion criteria

All exclusion criteria were pre-registered. Participants were excluded when reporting a severe neurological condition ($n = 14$) or English proficiency below B2 (i.e., good command/working knowledge; $n = 2$).

To check compliance with the questionnaires, four catch questions were presented during questionnaires, including two easy questions (e.g., 'Please answer "Not at all."') and two harder questions (e.g., 'In the past week, I (would have) wanted to eat mouldy food.', expected answer 'Disagree' or 'Definitely disagree'). Participants failing at least one easy question, or both harder questions were excluded ($n = 12$).

As task-based exclusion criteria, subjects rejecting all offers were excluded ($n = 0$). Participants had to have a clicking calibration score of at least seven, as values below would lead to challenges with just one mouse-click ($n = 4$). Subjects showing a large difference between minimum and maximum clicking speed (i.e., >3 SD) during calibration trials were excluded, as a misestimation of the calibration reference is likely ($n = 3$). Finally, subjects showing a large change in their clicking capacity (i.e., >3 SD) pre- to post-task were excluded, as it can be assumed the applied calibration was not valid during the task ($n = 1$). We also asked two open-answer questions after completion of the main task to monitor participants' self-reported task strategies as a way of assessing rule adherence.

## Analyses of effort-expenditure task data

### Model-agnostic analyses

Using the proportion of accepted challenges as the dependent variable, we investigated main effects of effort and reward levels and their interaction, using a repeated measures ANOVA of repeated measures. This approach accommodates the unbalanced design resulting from the implemented staircasing procedure.

### Model-based analyses

#### Model space

To model effort-based decision-making, we considered a model space of nine models. All models are variations of the economic decision-theory model, consisting of two basic equations. First, a cost function transforms costs and rewards associated with an action into a subjective value (SV):

$$SV = (\beta_R \cdot R) - (\beta_E \cdot E) \tag{1}$$

with $\beta_R$ and $\beta_E$ for reward and effort sensitivity, and $R$ and $E$ for reward and effort. Higher effort and reward sensitivity mean the SV is more strongly influenced by changes in effort and reward, respectively (*Figure 3B, C*). Hence, low effort and reward sensitivity mean the SV, and with that decision-making, is less guided by effort and reward offers, as would be in random decision-making.

This SV is then transformed to an acceptance probability by a softmax function:

$$p\,(accept) = \frac{1}{1 + e^{-(\alpha + SV)}} \tag{2}$$

with $p\,(accept)$ for the predicted acceptance probability and $\alpha$ for the intercept representing acceptance bias. A high acceptance bias means a subject has a bias, or tendency, to accept rather than reject effortful offers for reward (*Figure 3D*).

The models differed in two aspects. First, inclusion or exclusion of the free parameters reward sensitivity $(\beta_R)$ and acceptance bias $(\alpha)$. Second, the form of the cost function, which used either a linear function (proportional discounting at all effort levels), a parabolic function (increases at higher effort levels are discounted over-proportionally), or an exponential function (increases at lower effort-lower levels are discounted over-proportionally). See Mathematical definition of the model space for mathematical definitions of all models.

## Model fitting, checks, and comparisons

We took a hierarchical Bayesian approach to model fitting (*Ahn et al., 2011*), implemented with the CmdStan R interface (*Stan Development Team, 2021*), with Stan code adapted from hBayesDM (*Ahn et al., 2017*). Prior to model fitting, effort and reward levels were standardized for computational ease. All models were fit using Markov-Chain Monte Carlo (MCMC), with 2000 warm-up iterations and 6000 sampling iterations, by four chains. Model convergence and chain mixing were checked using numerical diagnostics of ESS and split R-hats, and by visually inspecting trace plots. We conducted parameter recoveries for all models, confirming their ability to meaningfully recover known parameters. Model performance was compared based on out-of-sample predictive accuracy using the LOOIC (lower is better) and ELPD (higher is better). The winning model was validated using posterior predictive checks, comparing model predictions to subject-wise observed choices.

## Test–retest reliability

We conducted an in-person study to validate the effort-expenditure task and assess the test–retest reliability of our computational modelling parameters. A sample of $N = 30$ participants was recruited and tested in two sessions, about 1 week apart. Test–retest reliability of task parameters was assessed by intra-class correlation coefficients, Pearson's correlation coefficients (estimated both after model fitting and by embedding a correlation matrix into the model fitting procedure), and by testing the predictive accuracy of parameter estimates across sessions. See Appendix 2 for full methods and results.

## Linking model parameters to outcome measures

To aid interpretability and comparability of effects, task parameters and questionnaire outcome measures were standardized to be between zero and one. Questionnaire measures resulting from the DARS, AES, and MEQ were additionally transformed to be interpretable with the same directionality within questionnaire groupings (i.e., for all psychiatric measures higher values are interpreted as higher symptom severity, for all circadian measures higher values are interpreted as later chronotype).

To investigate associations between effort-based decision-making and self-report questionnaires, we ran PLS regressions with questionnaire outcome measures predicting modelling parameters. PLS regression allows joint modelling of questionnaire measures, without issues due to expected multicollinearity between questionnaires. Following the best practice of model validation (*Dinga et al., 2019*), data were split into a training (75%) and a testing (25%) subset. The training data were used to obtain the optimal number of components, based on 10-fold-cross validation, and to train the model. The winning model's predictive performance was tested out-of-sample using the held-out testing data. Statistical significance of obtained effects (i.e., the predictive accuracy of the identified component and factor loadings) was assessed by permutation tests, probing the proportion of RMSEs indicating stronger or equally strong predictive accuracy under the null hypothesis.

To follow up on relationships suggested by the PLS regression, we performed Bayesian GLMs, adjusting for age and gender (male or female, imputing natal sex for non-binary participants, given low numbers).

## Comparing depressed and healthy subjects

We compared participants meeting criteria for a current MDD based on the M.I.N.I. (*Lecrubier et al., 1997*), to a subset of age- and gender-matched healthy controls (HCs, participants that did not meet criteria for current MDD). For computational sparsity, we only fit the three best-fitting models from the full sample. Models were fit separately to the MDD and HC groups, using the same methods and parameters described above. Bayesian GLMs were used to quantify evidence for associations between individual-level modelling parameters and group status. As we could not be certain whether we would obtain a large enough sample size of subjects meeting criteria for MDD, these analyses were exploratory.

## Investigating circadian effects

We used the two circadian rhythm questionnaires to determine participants' chronotypes. Early chronotype was defined as meeting criteria for 'morning types' on the MEQ (MEQ sum score > 58) (*Horne and Ostberg, 1976*) and having a midpoint of sleep on free days before 02:30 (*Roenneberg et al.,*

*2019*). Late chronotype was defined as meeting criteria for 'evening types' on the MEQ (MEQ sum score <42) and having a midpoint of sleep on free days after 05:30. Subjects not falling into either category are categorized as intermediate chronotypes and were not included in these analyses.

We used Bayesian age- and gender-controlled GLMs to investigate effects of chronotype, time-of-day (morning- vs. evening-testing), and their interaction on subject-wise mean task parameters estimates.

### Additional data collection

To improve the precision of estimated circadian effects on task parameters, we increased our sample size by conducting an additional pre-registered data collection (https://osf.io/y4fbe). We implemented a screening study comprising the MEQ (*Horne and Ostberg, 1976*) and MCTQ (*Roenneberg et al., 2003*). Taking the chronotyping approach described above, subjects with an early or late chronotype were identified. Early chronotypes were invited to take part in our study in the evening, late chronotypes in the morning.

We implemented a Bayesian stopping rule to inform our data collection process, taking the following steps. First, participants were screened in batches of 250, and eligible participants were invited to the study session. Next, data resulting from this additional data collection was joined with data resulting from the main data collection and Bayesian GLMs were re-run, as described above. If our precision target of any 95% HDI reaching a maximum width 0.20 was met, we stopped data collection. Was the precision target not met, we returned to step one, and another batch of 250 participants was screened. In any case, data collection would be terminated once 200 eligible participants had completed the main study session.

## Differentiating between the effects of neuropsychiatric symptoms and circadian measures on acceptance bias

To investigate how the effects of neuropsychiatric symptoms on acceptance bias (see Acceptance bias) relate to effects of chronotype and time-of-day on acceptance bias we conducted exploratory analyses. In the subsamples of participants with an early or late chronotype (including additionally collected data), we first ran Bayesian GLMs with neuropsychiatric questionnaire scores (SHAPS, DARS, and AES, respectively) predicting acceptance bias, controlling for age and gender. We next added an interaction term of chronotype and time-of-day into the GLMs, testing how this changes previously observed neuropsychiatric and circadian effects on acceptance bias. Finally, we conducted a model comparison using LOO, comparing between acceptance bias predicted by a neuropsychiatric questionnaire, acceptance bias predicted by chronotype and time-of-day, and acceptance bias predicted by a neuropsychiatric questionnaire and time-of-day (for each neuropsychiatric questionnaire, and controlling for age and gender).

## Acknowledgements

This study was funded by an AXA Research Fund Fellowship awarded to CLN (G102329) and the Medical Research Council (MC_UU_00030/12). CLN is funded by a Wellcome Career Development Award (226490/Z/22/Z). This research was supported by the NIHR Cambridge Biomedical Research Centre (BRC-1215-20014).

## Additional information

### Funding

| Funder | Grant reference number | Author |
| --- | --- | --- |
| AXA Research Fund | G102329 | Camilla L Nord |
| Medical Research Council | MC_UU_00030/12 | Camilla L Nord |
| Wellcome Trust | 10.35802/226490 | Camilla L Nord |

| Funder | Grant reference number | Author |
|---|---|---|
| NIHR Cambridge Biomedical Research Centre | BRC-1215- 20014 | Camilla L Nord |

The funders had no role in study design, data collection, and interpretation, or the decision to submit the work for publication. For the purpose of Open Access, the authors have applied a CC BY public copyright license to any Author Accepted Manuscript version arising from this submission.

### Author contributions

Sara Z Mehrhof, Conceptualization, Data curation, Software, Formal analysis, Validation, Investigation, Visualization, Methodology, Writing – original draft, Project administration, Writing – review and editing; Camilla L Nord, Conceptualization, Supervision, Funding acquisition, Investigation, Methodology, Writing – review and editing

### Author ORCIDs

Sara Z Mehrhof https://orcid.org/0000-0001-8267-3898
Camilla L Nord https://orcid.org/0000-0002-9281-3417

### Ethics

This study was approved by the University of Cambridge Human Biology Research Ethics Committee (HBREC.2020.40). Participants provided informed consent through an online form, complying with the University of Cambridge Human Biology Research Ethics Committee procedures for online studies.

Reviewer #1 (Public Review): https://doi.org/10.7554/eLife.96803.4.sa1
Reviewer #2 (Public Review): https://doi.org/10.7554/eLife.96803.4.sa2
Reviewer #3 (Public Review): https://doi.org/10.7554/eLife.96803.4.sa3
Author response https://doi.org/10.7554/eLife.96803.4.sa4

## Additional files

### Supplementary files
MDAR checklist

### Data availability

The anonymous processed data, the R and Stan code to reproduce all simulations, analyses, and visualizations, and all experimental materials are publicly available on GitHub (https://github.com/smehrhof/2024_effort_study; copy archived at *Mehrhof, 2025*) and published on Zenodo (https://doi.org/10.5281/zenodo.15068968).

The following dataset was generated:

| Author(s) | Year | Dataset title | Dataset URL | Database and Identifier |
|---|---|---|---|---|
| Mehrhof SZ, Nord CL | 2025 | smehrhof/effort-study: eLife Publication 22/03/2025 | https://doi.org/10.5281/zenodo.15068968 | Zenodo, 10.5281/zenodo.15068968 |

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

# Appendix 1

## 1 Computational modelling

### 1.1 Mathematical definition of the model space

**Appendix 1—table 1.** Mathematical definition of the models included in our model space.

| Model | Cost function | Softmax function |
|---|---|---|
| Linear model 1 | $SV = (R) - (\beta_E \cdot E)$ | $p\left(accept\right) = \dfrac{1}{1 + e^{-\left(\alpha + SV\right)}}$ |
| Linear model 2 | $SV = (\beta_R \cdot R) - (\beta_E \cdot E)$ | $p\left(accept\right) = \dfrac{1}{1 + e^{-SV}}$ |
| Linear model 3 | $SV = (\beta_R \cdot R) - (\beta_E \cdot E)$ | $p\left(accept\right) = \dfrac{1}{1 + e^{-\left(\alpha + SV\right)}}$ |
| Parabolic model 1 | $SV = (R) - \left(\beta_E \cdot E^2\right)$ | $p\left(accept\right) = \dfrac{1}{1 + e^{-\left(\alpha + SV\right)}}$ |
| Parabolic model 2 | $SV = (\beta_R \cdot R) - \left(\beta_E \cdot E^2\right)$ | $p\left(accept\right) = \dfrac{1}{1 + e^{-SV}}$ |
| Parabolic model 3 | $SV = (\beta_R \cdot R) - \left(\beta_E \cdot E^2\right)$ | $p\left(accept\right) = \dfrac{1}{1 + e^{-\left(\alpha + SV\right)}}$ |
| Exponential model 1 | $SV = (R) \cdot e^{\left(-\beta_E \cdot E\right)}$ | $p\left(accept\right) = \dfrac{1}{1 + e^{-5 \cdot \left(\alpha + SV\right)}}$ |
| Exponential model 2 | $SV = (\beta_R \cdot R) \cdot e^{\left(-\beta_E \cdot E\right)}$ | $p\left(accept\right) = \dfrac{1}{1 + e^{-5 \cdot \left(-0.5 + SV\right)}}$ |
| Exponential model 3 | $SV = (\beta_R \cdot R) \cdot e^{\left(-\beta_E \cdot E\right)}$ | $p\left(accept\right) = \dfrac{1}{1 + e^{-5 \cdot \left(\alpha + SV\right)}}$ |

### 1.2 Model validation

Parameter recoveries were performed to ensure parameter estimates obtained from all models are meaningful. For each model, sets of parameter values were sampled from uniform distributions bound to the respective parameter ranges. Task data were then simulated for $n = 500$ agents, using the respective parameters and modelling equations. The resulting simulated data were used for model fitting and resulting 'recovered' posterior parameter estimates compared to the underlying parameter values. For all models, underlying parameters correlated highly with the recovered mean parameter estimates (*Appendix 1—table 2*). For the winning model (full parabolic model), relations between underlying and recovered parameters are additionally visualized in *Appendix 1—figure 1A–C*. Importantly, the modelling procedure did not introduce any spurious correlations between free parameters (*Appendix 1—figure 1D*).

**Appendix 1—table 2.** Pearson's correlations between underlying parameters and recovered mean parameter estimates for all models included in the model space.

| | Linear models | | | Parabolic models | | | Exponential models | | |
|---|---|---|---|---|---|---|---|---|---|
| | 1 | 2 | 3 | 1 | 2 | 3 | 1 | 2 | 3 |
| $\beta_E$ | 0.811 | 0.927 | 0.816 | 0.836 | 0.919 | 0.838 | 0.926 | 0.882 | 0.802 |
| $\beta_R$ | - | 0.919 | 0.841 | - | 0.928 | 0.851 | - | 0.898 | 0.891 |
| $\alpha$ | 0.965 | - | 0.932 | 0.978 | - | 0.932 | 0.950 | - | 0.904 |

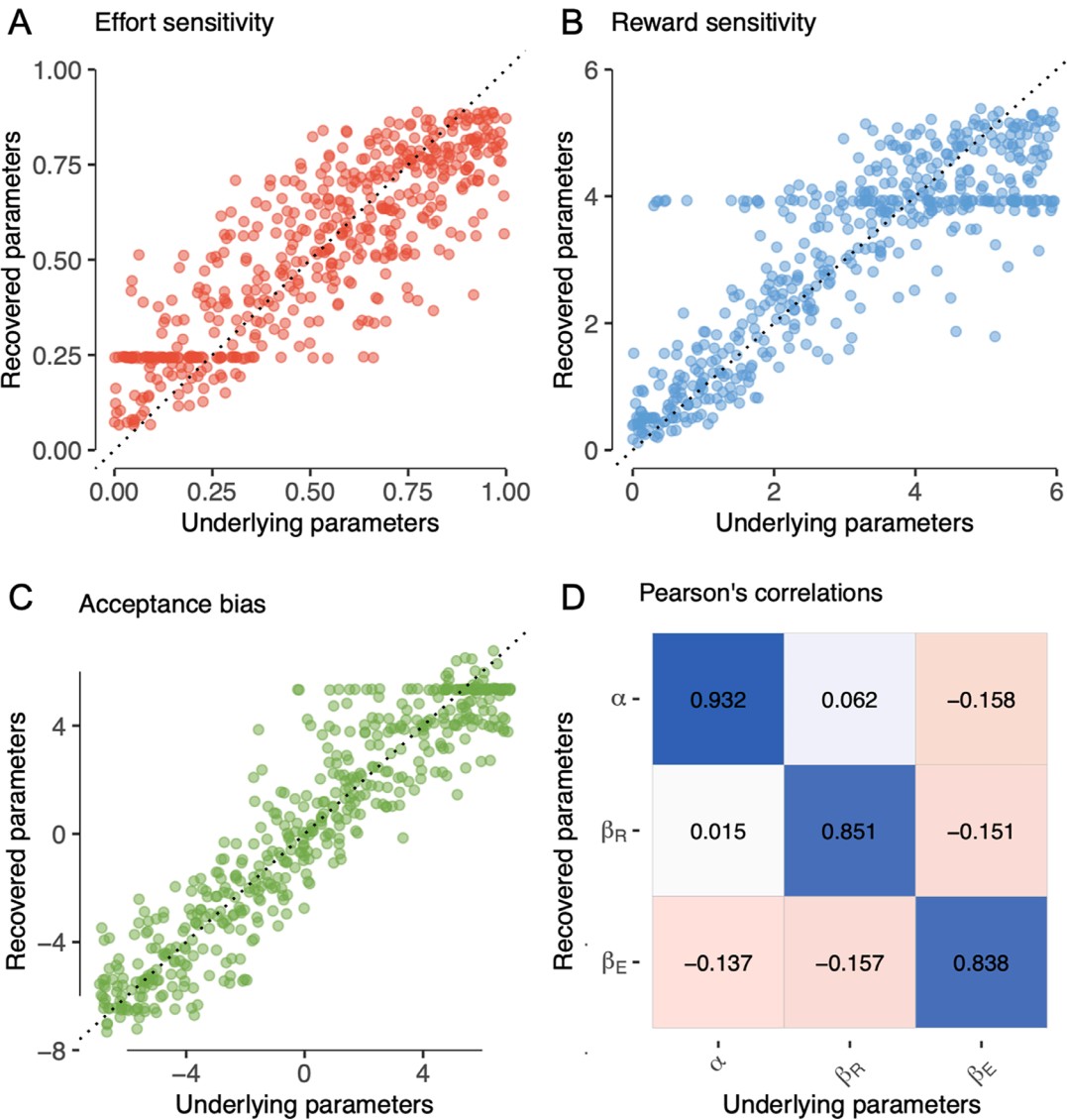

**Appendix 1—figure 1.** Parameter recovery. (**A–C**) Comparison between underlying parameters and recovered mean parameter estimates for the three free parameters of the full parabolic model. (**D**) Pearson's correlations between all underlying and recovered parameters for the full parabolic model.

## 1.2.1 Parameter recoveries including inverse temperature

In the process of task and model space development, we also considered models incorporating an inverse temperature paramater. To this end, we conducted parameter recoveries for four models, defined in ***Appendix 1—table 3***.

Parameter recoveries indicated that, parameters can be recovered reliably in model 1, which includes only effort sensitivity ($\beta_E$) and inverse temperature ($\tau$) as free parameters (on-diagonal correlations: $0.98 > r > 0.89$, off-diagonal correlations: $0.04 > |r| > 0.004$). However, as a reward sensitivity parameter is added to the model (model 2), parameter recovery seems to be compromised, as parameters are estimated less accurately (on-diagonal correlations: $0.80 > r > 0.68$), and spurious correlations between parameters emerge (off-diagonal correlations: $0.40 > |r| > 0.17$). This issue remains when acceptance bias is added to the model (model 4; on-diagonal correlations: $0.90 > r > 0.65$; off-diagonal correlations: $0.28 > |r| > 0.03$), but not when inverse temperature is modelled with effort sensitivity and acceptance bias, but not reward sensitivity (model 3; on-diagonal correlations: $0.96 > r > 0.73$; off-diagonal correlations: $0.05 > |r| > 0.003$).

As our pre-registered hypotheses related to the reward sensitivity parameter, we opted to include models with the reward sensitivity parameter rather than the inverse temperature parameter in our model space.

**Appendix 1—table 3.** Mathematical definition of models included an inverse temperature parameter.

| Model | Cost function | Softmax function |
|---|---|---|
| 1 | $SV = (R) - \left(\beta_E \cdot E^2\right)$ | $p\left(accept\right) = \frac{1}{1+e^{-\tau(SV)}}$ |
| 2 | $SV = (\boldsymbol{\beta_R} \cdot R) - \left(\boldsymbol{\beta_E} \cdot E^2\right)$ | $p\left(accept\right) = \frac{1}{1+e^{-\tau(SV)}}$ |
| 3 | $SV = (R) - \left(\boldsymbol{\beta_E} \cdot E^2\right)$ | $p\left(accept\right) = \frac{1}{1+e^{-\tau(\alpha+SV)}}$ |
| 4 | $SV = (\boldsymbol{\beta_R} \cdot R) - \left(\boldsymbol{\beta_E} \cdot E^2\right)$ | $p\left(accept\right) = \frac{1}{1+e^{-\tau(\alpha+SV)}}$ |

## 1.3 Group and subject-wise parameter estimates

As described in the main manuscript, we took a hierarchical Bayesian approach to model fitting (*Ahn et al., 2011*), using Markov-Chain Monte Carlo (MCMC) sampling. Hence, we obtained posterior parameter distributions for each parameter, on both the subject and group level. Analyses presented in the main manuscript are based on the mean for each subject-wise parameter estimate distribution. The resulting parameter estimates—both subject-wise and on a group-level—are visualized in *Appendix 1—figure 2*.

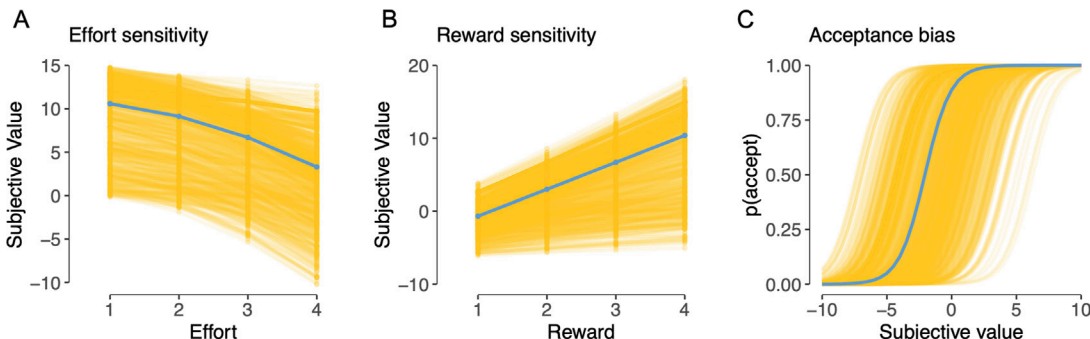

**Appendix 1—figure 2.** Parameter estimates. (**A–C**) Visualization of individual-level (yellow) and group-level (blue) model parameter estimates for effort sensitivity (**A**), reward sensitivity (**B**), and acceptance bias (**C**).

## Appendix 2

### 2 Test–retest reliability

For computational modelling parameters to successfully contribute to research advancing our understanding of mechanisms underlying mental health and disorder, as well as to transform such knowledge into personalized treatment approaches, it is pivotal to ensure reliability of the used measures. The reliability with which a measure captures individual characteristics ultimately sets an upper limit on its usefulness in detecting differences between groups, relationships to other measures, and intervention effects. We assessed the test–retest reliability of the novel effort-expenditure task in a separate in-person sample.

### 2.1 Methods

#### 2.1.1 Sample

Thirty-three participants were recruited from S.O.N.A. and through advertisements in Cambridge Colleges. Three subjects had to be excluded due to failure during the task calibration. The final sample ($N = 30$) consisted of 17 female and 13 male subjects with an average age of $M = 48.1$ years (SD = 21.54). All subjects were native English speakers, none reported neurological conditions and four subjects reported current and/or past psychiatric disorders (depression, anxiety, obsessive–compulsive disorder, and personality disorder).

#### 2.1.2 Study procedure

All participants completed two testing sessions. In the first session, demographic data were collected, followed by the effort-expenditure task and the battery of self-report questionnaires. In the second session, subjects only completed the effort task. The two testing sessions were at least 6 days apart and 9 days at most, with an average difference of 6.93 days (SD = 0.78). Time-of-day at testing was aimed to be held constant between testing sessions, with the average time difference between starting times of the task being 21.53 min (SD = 30.44, min = 0, max = 119).

#### 2.1.3 Analyses

All computational models defined in our model space were fit separately to the task data of each testing session using the same methods and parameters as described for the main sample in the main manuscript. Test–retest analyses were performed on the full parabolic model, given model comparison in our main sample identified this as the winning model.

Intra-class correlation coefficients (ICCs) were used to assess test–retest reliability using the ratio of intra- to inter-individual variability. We used two-way mixed effects ICCs considering consistency (reflecting rank order). ICCs below 0.4 indicate that a measure is not reliable, ICCs between 0.4 and 0.75 indicate moderate to good reliability and ICCs above 0.75 indicate excellent reliability (*Fleiss, 2011*).

To assess correlations between model parameters from the two sessions, we first calculated Pearson's correlation coefficients between mean posterior parameter estimates resulting from the separate model fitting procedures. Next, we re-fit the model jointly for both testing sessions, while embedding a correlation matrix into the model (*Haines et al., 2020*). Thereby, the full posterior distributions of the model parameters are fed into the calculation of correlation coefficients, rather than solely point estimates. This offers multiple benefits of Bayesian inference: Uncertainty around parameter estimates can be incorporated into the correlation estimation and Bayesian priors can be set over possible values of the correlation matrix. Note that while this model fitting procedure fits the data of both testing sessions simultaneously, separate hyper-parameters are used for the different sessions. Therefore, shrinkage cannot bias the reliability estimates.

Finally, we also made use of the predictive property of the computational modelling approach in the assessment of parameter reliability. If parameters are reliable, the estimates resulting from modelling the data from one session should be able to predict subjects' choices in the other session better than chance (*Mkrtchian et al., 2023*). To test this, we calculated the model-predicted choices for each subject and trial and compared this to the observed choices in the respective other session.

Due to our hierarchical Bayesian modelling approach, individual model parameters are subject to shrinkage. To test whether the predictive property of individual parameters is solely due to shrinkage

we repeated the procedure using hyper-parameters to make model predictions and then compared the resulting predictive accuracy to that of individual parameters.

## 2.2 Results

### 2.2.1 Descriptive task statistics

Data from both sessions reproduced the expected effect of effort discounting. Mixed-effects ANOVAs of repeated measures confirmed significant main effects of effort (session 1: $F(1,447) = 128.22$, p < 0.001; session 2: $F(1,447) = 129.42$, p < 0.001) and reward (session 1: $F(1,447) = 56.49$, p < 0.001; session 2: reward: $F(1,447) = 46.51$, p < 0.001), as well as an interaction effect (session 1: $F(1,447) = 52.74$, p < 0.001; session 2: $F(1,447) = 49.13$, p < 0.001) for both testing sessions. In post hoc ANOVAs, the main effect of effort remained significant at all reward levels (at p < 0.05) for both sessions and the main effect of reward remained significant at all effort levels (at p < 0.05), except at the lowest effort level in the first session, and at the two lowest reward levels in the second session.

### 2.2.2 Computational modelling

All models converged well for both testing sessions. For both sessions 1 and 2, the full parabolic model was the winning model, based on both the leave-one-out information criterion (LOO) and the expected log predictive density (ELPD) (***Appendix 2—figure 1A***).

### 2.2.3 Test–retest reliability

The ICC for effort sensitivity showed best test–retest reliability (ICC(C,1) = 0.797, p < 0.001). Reliability for reward sensitivity (ICC(C,1) = 0.459, p = 0.0047) and acceptance bias (ICC(C,1) = 0.463, p = 0.0043), was moderate (***Appendix 2—figure 1B***).

Correlations between point estimates of the modelling parameters across testing sessions were very strong for effort sensitivity ($r = 0.803$, p < 0.01), and moderate for reward sensitivity ($r = 0.467$, p < 0.01) and acceptance bias ($r = 0.517$, p < 0.01). Correlations derived from the embedded correlation matrix and therefore considering the full parameter estimate distribution resulted in slightly higher estimates (effort sensitivity: $r = 0.867$; 95% highest density interval (HDI) = [0.667–0.999]; reward sensitivity: $r = 0.550$; 95% HDI = [0.163–0.900]; acceptance bias: $r = 0.585$; 95% HDI = [0.184–0.927]).

Subject-wise parameter estimates could reliably predict individual trial-wise choice data across trials (***Appendix 2—figure 1C***). Session 1 parameter estimates predicted session 2 choice data significantly better than chance ($t(1919) = 46.819$, p < 0.001), as did session 2 parameters for session 1 choice data ($t(1919) = 47.106$, p < 0.001). When considering predictive accuracy of group-level parameter estimates, both session's parameter estimates outperformed chance (session 1 predicting session 2: $t(1919) = 33.485$, p < 0.001; session 2 predicting session 1: $t(1919) = 30.291$, p < 0.001). Comparing predictive accuracy of subject-wise parameter estimates to group-level estimates, the subject-level outperformed the group level for both session 1 predicting session 2 ($t(3760.7) = 4.951$, p < 0.001), and session 2 predicting session 1 ($t(3674.2) = 7.426$, p < 0.001).

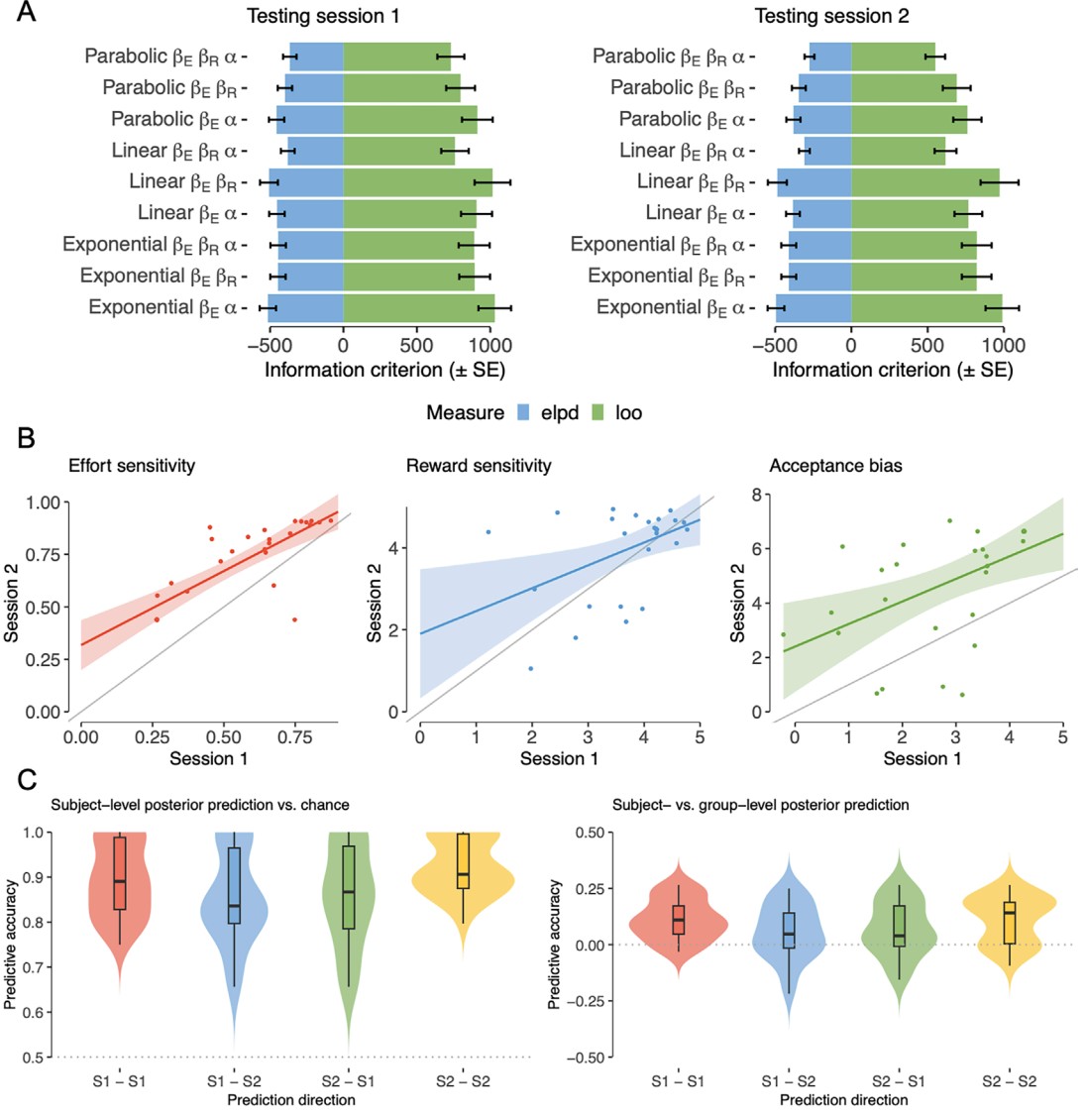

**Appendix 2—figure 1.** Computational modelling and test–retest reliability. (**A**) Model comparison for each testing session based on the leave-one-out information criterion (LOO) and expected log predictive density (ELPD). Error bars indicate standard errors (n = 30). (**B**) Subject-wise parameter estimates compared between testing sessions. (**C**) Predictive accuracy against chance (left) and group-level parameters (right; values >0 indicate better performance of subject-level compared to group-level parameters). Labels s1s2 (and s2s1) indicate session 1 (session 2) parameters predicting session 2 (session 1) data, s1s1 (and s2s2) indicate session 1 (session 2) parameters predicting session 1 (session 2) data.

## Appendix 3

### 3 Model-agnostic task measures relating to questionnaires

#### 3.1 Proportion of accepted trials

To explore the relationship between model-agnostic task measures to questionnaire measures of neuropsychiatric symptoms, we conducted Bayesian GLMs, with the proportion of accepted trials predicted by SHAPS scores, controlling for age and gender. The proportion of accepted trials averaged across effort and reward levels was predicted by the Snaith–Hamilton Pleasure Scale (SHAPS) sum scores ($M = -0.07$; 95% HDI=[$-0.12,-0.03$]) and the Apathy Evaluation Scale (AES) sum scores ($M = -0.05$; 95% HDI = [$-0.10,-0.002$]). Note that this was not driven only by higher effort levels; even confining data to the lowest two effort levels, SHAPS has a predictive value for the proportion of accepted trials: $M = -0.05$; 95% HDI = [$-0.07,-0.02$].

A visualization of model-agnostic task measures relating to symptoms is given in *Appendix 3—figure 1*, comparing subgroups of participants scoring in the highest and lowest quartile on the SHAPS. This shows that participants with a high SHAPS score (i.e., more pronounced anhedonia) are less likely to accept offers than those with a low SHAPS score (*Appendix 3—figure 1A*). Due to the implemented staircasing procedure, group differences can also be seen in the effort–reward combinations offered per trial. While for both groups, the staircasing procedure seems to devolve towards high effort—low reward offers, this is more pronounced in the subgroup of participants with a lower SHAPS score (*Appendix 3—figure 1B*).

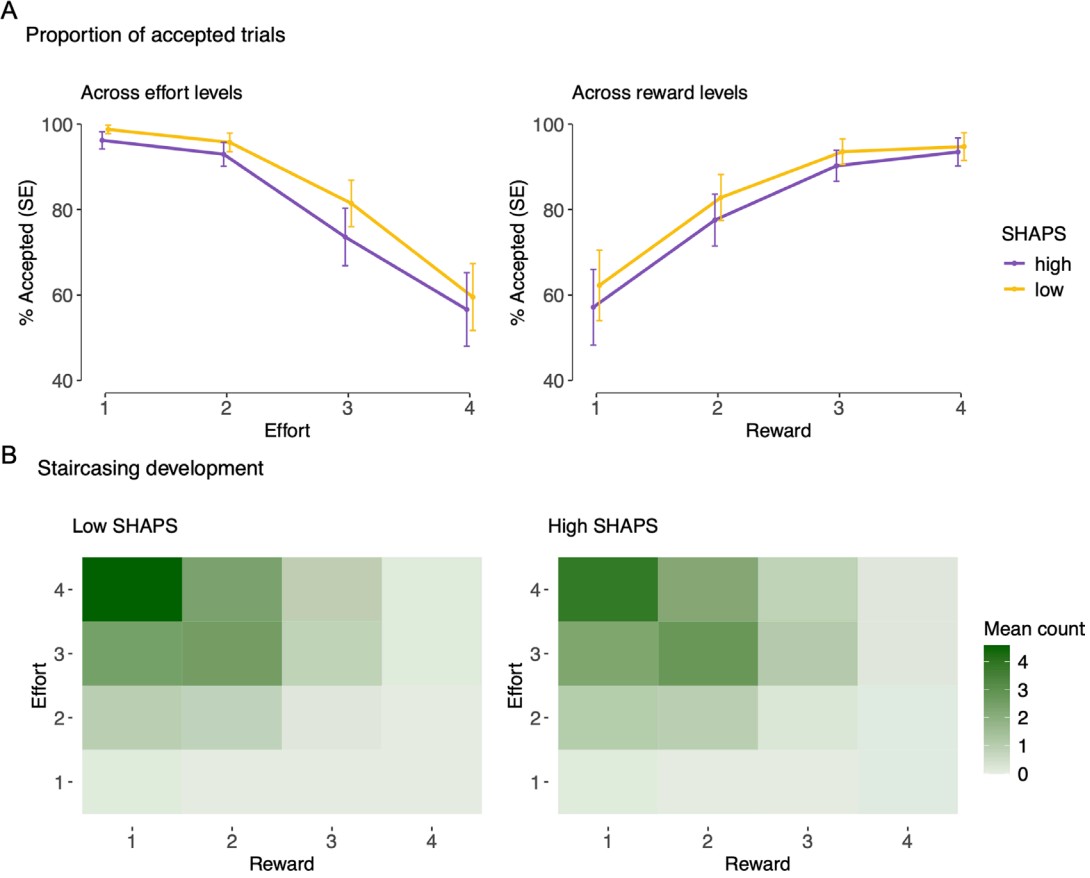

**Appendix 3—figure 1.** Model-agnostic task measures relation to anhedonia. (**A**) Comparing the proportion of accepted trials across effort (right) and reward (left) levels in subsamples of participants scoring in the highest and lowest SHAPS quartile. Error bars indicate standard errors (n = 479). (**B**) Distribution of effort–reward combinations, averaged across the final trial of 16 staircases.

## 3.2 Proportion of accepted but failed trials

For each participant, we computed the proportion of trial in which an offer was accepted, but the required effort then not fulfilled (i.e., failed trials). There was no relationship between average proportion of accepted but failed trials and SHAPS score (controlling for age and gender): $M = 0.01$, 95% HDI = [–0.001,0.02]. However, there are intentionally few accepted but failed trials ($M = 1.3\%$ trials failed, SD = 3.50). This results from several task design characteristics aimed at preventing subjects from failing accepted trials, to avoid confounding of effort discounting with risk discounting.

## 3.3 Exertion of 'extra effort'

We also explored the extent to which participants went 'above and beyond' the target in accepted trials. Specifically, considering only accepted and succeeded trials, we computed the factor by which the required number of clicks was exceeded (i.e., if a subject clicked 15 times when 10 clicks were required the factor would be 1.3), averaging across effort and reward level. We then conducted a Bayesian GLM to test whether this subject-wise click-exceedance measure can be predicted by apathy or anhedonia, controlling for age and gender. We found neither the SHAPS ($M = -0.14$, 95% HDI = [–0.43,0.17]) nor the AES ($M = 0.07$, 95% HDI = [–0.26,0.41]) had a predictive value for the amount to which subjects exert 'extra effort'.

## 4 Subject-reported decision-making process

After completing the task, subjects were asked whether they used "any strategy to facilitate the game". While this question was initially included to monitor any self-reported cheating strategies, a large subset of subjects (425 subjects, 44.36%) understood this question as a prompt to report their experience of the decision-making process during the effort-expenditure task. Some examples of participants' reports are given below:

- "Only accepting the challenge if the points were equal to or greater than the effort level."
- "I decided whether the points were worth the effort."
- "I tended to reject if the reward was only 2."
- "Measure of how many points to gain against effort and how tired my hand felt."
- "As the game progressed I decided what ratio of effort to reward that I would tolerate."
- "I rejected the higher difficulties which had low points."
- "Was it worth the effort for the points?"
- "I didn't do the higher effort challenges."

