## [Editor Report · eLife assessment]

This **important** study provides **convincing** evidence that both psychiatric dimensions (e.g. anhedonia, apathy, or depression) and chronotype (i.e., being a morning or evening person) influence effort-based decision-making. This is of importance to researchers and clinicians alike, who may make inferences about behaviour and cognition without taking into account whether the individual may be tested or observed out-of-sync with their phenotype. The current study can serve as a starting point for more targeted investigation of the relationship between chronotype, altered decision making and psychiatric illness.

---

## [Referee Report · Reviewer #1 (Public Review)]

Summary:

This study uses an online cognitive task to assess how reward and effort are integrated in a motivated decision-making task. In particular the authors were looking to explore how neuropsychiatric symptoms, in particular, apathy and anhedonia, and circadian rhythms affect behavior in this task. Amongst many results, they found that choice bias (the degree to which integrated reward and effort affect decisions) is reduced in individuals with greater neuropsychiatric symptoms, and late chronotypes (being an 'evening person').

Strengths:

The authors recruited participants to perform the cognitive task both in and out of sync with their chronotypes, allowing for the important insight that individuals with late chronotypes show a more reduced choice bias when tested in the morning.

Overall, this is a well-designed and controlled online experimental study. The modelling approach is robust, with care being taken to both perform and explain to the readers the various tests used to ensure the models allow the authors to sufficiently test their hypotheses.

Weaknesses:

This study was not designed to test the interactions of neuropsychiatric symptoms and chronotypes on decision making, and thus can only make preliminary suggestions regarding how symptoms, chronotypes and time-of-assessment interact.

---

## [Referee Report · Reviewer #2 (Public Review)]

Summary:

The study combines computational modeling of choice behavior with an economic, effort-based decision-making task to assess how willingness to exert physical effort for a reward varies as a function of individual differences in apathy and anhedonia, or depression, as well as chronotype. They find an overall reduction in effort selection that scales with apathy, anhedonia and depression. They also find that later chronotypes are less likely to choose effort than earlier chronotypes and, interestingly, an interaction whereby later chronotypes are especially unwilling to exert effort in the morning versus the evening.

Strengths:

This study uses state-of-the-art tools for model fitting and validation and regression methods which rule out multicollinearity among symptom measures and Bayesian methods which estimate effects and uncertainty about those estimates. The replication of results across two different kinds of samples is another strength. Finally, the study provides new information about the effects not only of chronotype but also chronotype by timepoint interactions which are previously unknown in the subfield of effort-based decision-making.

Weaknesses:

The study has few weaknesses. The biggest drawback is that it does not provide evidence for the idea that a match between chronotype and delay matters is especially relevant for people with depression or continuous measures like anhedonia and apathy. It is unclear whether disorders further interact with chronotype and time of day to determine a bias against effort. On the other hand, the study does provide evidence that future studies should consider such interactions when examining questions about effort expenditure in psychiatric disorders.

---

## [Referee Report · Reviewer #3 (Public Review)]

Summary:

In this manuscript, Mehrhof and Nord study a large dataset of participants collected online (n=958 after exclusions) who performed a simple effort-based choice task. They report that the level of effort and reward influence choices in a way that is expected from prior work. They then relate choice preferences to neuropsychiatric syndromes and, in a smaller sample (n<200), to people's circadian preferences, i.e., whether they are a morning-preferring or evening-preferring chronotype. They find relationships between the choice bias (a model parameter capturing the likelihood to accept effort-reward challenges, like an intercept) and anhedonia and apathy, as well as chronotype. People with higher anhedonia and apathy and an evening chronotype are less likely to accept challenges (more negative choice bias). People with an evening chronotype are also more reward sensitive and more likely to accept challenges in the evening, compared to the morning.

Strengths:

This is an interesting and well-written manuscript which replicates some known results and introduces a new consideration related to chronotype relationships which have not been explored before. It uses a large sample size and includes analyses related to transdiagnostic as well as diagnostic criteria.

Weaknesses:

The authors do not explore how chronotype and depression are related (does one mediate the effect of the other etc). Both variables are included in the same model in the revised article now which is a great improvement, but it also means psychopathology and circadian rhythms are treated as distinct phenomena and their relationship in predicting effort-reward preferences is not examined.

---

## [Author Response]

The following is the authors’ response to the previous reviews.

**Public Reviews:**

**Reviewer #1 (Public Review):**
Summary:This study uses an online cognitive task to assess how reward and effort are integrated in a motivated decision-making task. In particular the authors were looking to explore how neuropsychiatric symptoms, in particular, apathy and anhedonia, and circadian rhythms affect behavior in this task. Amongst many results, they found that choice bias (the degree to which integrated reward and effort affect decisions) is reduced in individuals with greater neuropsychiatric symptoms, and late chronotypes (being an 'evening person').Strengths:The authors recruited participants to perform the cognitive task both in and out of sync with their chronotypes, allowing for the important insight that individuals with late chronotypes show a more reduced choice bias when tested in the morning.Overall, this is a well-designed and controlled online experimental study. The modelling approach is robust, with care being taken to both perform and explain to the readers the various tests used to ensure the models allow the authors to sufficiently test their hypotheses.Weaknesses:This study was not designed to test the interactions of neuropsychiatric symptoms and chronotypes on decision making, and thus can only make preliminary suggestions regarding how symptoms, chronotypes and time-of-assessment interact.
**Reviewer #2 (Public Review):**
Summary:The study combines computational modeling of choice behavior with an economic, effort-based decision-making task to assess how willingness to exert physical effort for a reward varies as a function of individual differences in apathy and anhedonia, or depression, as well as chronotype. They find an overall reduction in effort selection that scales with apathy, anhedonia and depression. They also find that later chronotypes are less likely to choose effort than earlier chronotypes and, interestingly, an interaction whereby later chronotypes are especially unwilling to exert effort in the morning versus the evening.Strengths:This study uses state-of-the-art tools for model fitting and validation and regression methods which rule out multicollinearity among symptom measures and Bayesian methods which estimate effects and uncertainty about those estimates. The replication of results across two different kinds of samples is another strength. Finally, the study provides new information about the effects not only of chronotype but also chronotype by timepoint interactions which are previously unknown in the subfield of effort-based decision-making.Weaknesses:The study has few weaknesses. The biggest drawback is that it does not provide evidence for the idea that a match between chronotype and delay matters is especially relevant for people with depression or continuous measures like anhedonia and apathy. It is unclear whether disorders further interact with chronotype and time of day to determine a bias against effort. On the other hand, the study does provide evidence that future studies should consider such interactions when examining questions about effort expenditure in psychiatric disorders.
**Reviewer #3 (Public Review):**
Summary:In this manuscript, Mehrhof and Nord study a large dataset of participants collected online (n=958 after exclusions) who performed a simple effort-based choice task. They report that the level of effort and reward influence choices in a way that is expected from prior work. They then relate choice preferences to neuropsychiatric syndromes and, in a smaller sample (n<200), to people's circadian preferences, i.e., whether they are a morning-preferring or evening-preferring chronotype. They find relationships between the choice bias (a model parameter capturing the likelihood to accept effort-reward challenges, like an intercept) and anhedonia and apathy, as well as chronotype. People with higher anhedonia and apathy and an evening chronotype are less likely to accept challenges (more negative choice bias). People with an evening chronotype are also more reward sensitive and more likely to accept challenges in the evening, compared to the morning.Strengths:This is an interesting and well-written manuscript which replicates some known results and introduces a new consideration related to chronotype relationships which have not been explored before. It uses a large sample size and includes analyses related to transdiagnostic as well as diagnostic criteria.Weaknesses:The authors do not explore how chronotype and depression are related (does one mediate the effect of the other etc). Both variables are included in the same model in the revised article now which is a great improvement, but it also means psychopathology and circadian rhythms are treated as distinct phenomena and their relationship in predicting effort-reward preferences is not examined.
**Recommendations for the authors:**

**Reviewer #3 (Recommendations For The Authors):**
Two points in response to changes the authors made:(1) "motivational tendency" is in our opinion not an improved phrase over "choice bias". A paper by Jon Roiser calls it "overall bias to accept effortful challenges" (but that's maybe too long?)

We thank the reviewer for their suggestion of renaming our computational parameter and agree it would be of value to introduce and label this parameter in line with other work, improving consistency across the literature. Hence, we have updated our manuscript and now introduce the parameter as bias to accept effortful challenges for reward and refer to the parameter as acceptance bias thereafter.

We have updated this nomenclature throughout the manuscript text, figures and supplement.

(2) The new title "Both neuropsychiatric symptoms and circadian rhythm alter effort-based decision-making" sounds slightly causal (as would be the case in a longitudinal or intervention study). Maybe instead the authors could use "are associated with" or similar?

We agree with the reviewers that our current title could be interpreted in a causal manner. We have updated our title to now read A common alteration in effort-based decision-making in apathy, anhedonia, and late circadian rhythm.